# Image-based modeling of kidney branching morphogenesis reveals GDNF-RET based Turing-type mechanism and pattern-modulating WNT11 feedback

Denis Menshykau[1,2], Odyssé Michos [1,2,3], Christine Lang[1,2], Lisa Conrad[1,2], Andrew P. McMahon[3] & Dagmar Iber [1,2]

Branching patterns and regulatory networks differ between branched organs. It has remained unclear whether a common regulatory mechanism exists and how organ-specific patterns can emerge. Of all previously proposed signalling-based mechanisms, only a ligand-receptor-based Turing mechanism based on FGF10 and SHH quantitatively recapitulates the lung branching patterns. We now show that a GDNF-dependent ligand-receptor-based Turing mechanism quantitatively recapitulates branching of cultured wildtype and mutant ureteric buds, and achieves similar branching patterns when directing domain outgrowth in silico. We further predict and confirm experimentally that the kidney-specific positive feedback between WNT11 and GDNF permits the dense packing of ureteric tips. We conclude that the ligand-receptor based Turing mechanism presents a common regulatory mechanism for lungs and kidneys, despite the differences in the molecular implementation. Given its flexibility and robustness, we expect that the ligand-receptor-based Turing mechanism constitutes a likely general mechanism to guide branching morphogenesis and other symmetry breaks during organogenesis.

[1] Department for Biosystems Science and Engineering, ETH Zurich, Mattenstrasse 26, 4058 Basel, Switzerland. [2] Swiss Institute of Bioinformatics, Mattenstrasse 26, 4058 Basel, Switzerland. [3] Department of Stem Cell Biology and Regenerative Medicine, Eli and Edythe Broad Center for Regenerative Medicine and Stem Cell Research, University of Southern California Keck School of Medicine, Los Angeles, CA 90089, USA. Correspondence and requests for materials should be addressed to D.M. (email: denis.menshykau@gmail.com) or to D.I. (email: dagmar.iber@bsse.ethz.ch)

Branched epithelial trees are found in many organs, and the capacity of lungs, glands, and kidneys to perform their physiological function depends on the emergence of the correct branching structure[1,2]. The diversity of shapes and functions makes branching morphogenesis an excellent system to establish both general principles of morphogenesis and mechanisms responsible for the shaping of a particular organ[3–5]. The overall subtree organisation in mammary glands and kidneys has been proposed to follow from a density-dependent termination of branching[6], though key assumptions have been challenged[7]. How the local branching rules, i.e. the distance and angles between new branches, arise remains an open question.

The first rounds of lung and kidney branching are stereotyped[8,9]. A deterministic rather than a stochastic process must therefore control branching morphogenesis, and signalling plays a key role[3]. Thus, FGF10/FGFR2b and GDNF/RET signalling concentrates at the tips of lung and ureteric buds, respectively[10,11] (Supplementary Figures 1, 2), and induces epithelial outgrowth[12–14]. In the absence of signalling, new branches cannot form[1,2,15–21]. Accordingly, the correct model needs to explain how growth factor signalling becomes concentrated at the points of outgrowth, in spite of uniform ligand production in the adjacent mesenchyme[22–24]. Many other signalling pathways modulate the observed branching pattern, but in their absence branching still proceeds.

The pathways that are necessary for branching morphogenesis, i.e. the "core" regulatory networks, share three common motifs: (i) FGF10 and GDNF are dimers, which likely interact cooperatively with their receptors[25,26] (Fig. 1a, black arrows), (ii) ligand–receptor signalling triggers an upregulation of receptor abundance[11,27] (Fig. 1a, green arrow), and (iii) Fgf10 and Gdnf, are expressed in the mesenchyme, while their receptors, FgfrIIb and Ret, are expressed in the epithelium[28–31] (Fig. 1b). We have previously shown that the first two properties, together with the higher diffusion coefficient of extracellular ligands compared to their membrane receptors, are necessary to result in patterning via a Turing mechanism, and that the third property ensures pattern robustness in spite of noisy initial conditions[32–34]. FGF10 engages in a negative feedback with Sonic Hedgehog (SHH) signalling (Fig. 1c)[15,35,36]. SHH and its receptor PTCH also meet the conditions of a ligand–receptor-based Turing mechanism and the negative feedback between FGF10 and SHH massively increases the size of the parameter space for which Turing patterns are observed[34]. We showed that the ligand–receptor-based Turing mechanisms generates FGF10/SHH and GDNF signalling

patterns that qualitatively match the experimentally observed gene expression and branching patterns in wildtype and mutant developing lungs and kidneys[32,33,37].

Other signalling mechanisms have been proposed to explain the control of lung branching[3]. Given that Fgf10 expression is strongest furthest away from the epithelium, differences in mesenchyme thickness have been proposed to generate the observed patterns[30,38,39]. Alternatively, the curvature of the bud has been proposed to induce a concentration profile[40,41]. By combining imaging data with computational modelling, we showed that of all previously proposed signalling-based mechanisms only a ligand–receptor-based Turing-type mechanism in combination with a tissue-specific expression of ligands and receptors[33,34,42] correctly recapitulates the experimentally observed areas of growth during lung branching morphogenesis[43]. Moreover, unlike the other proposed mechanisms, a ligand–receptor-based Turing mechanism can also explain how patterns and thus branches can still emerge when Fgf10 is ubiquitously expressed[24].

Recently, a coupling of morphogen dynamics and morphogen-induced shape changes has been shown to result in patterning[44]. Here, the shape changes had to result in locally enhanced morphogen production. The latter would not fit with the situation in lungs and kidneys where the morphogens are produced in a different tissue (mesenchyme) from the one that deforms (epithelium). Moreover, tissue stretching as predicted from mechanical models did not coincide with measured branch points[45]. While more complicated feedback architectures that involve morphogens and tissue mechanics could be explored, we note that the ligand–receptor-based Turing mechanism represents a parsimonious mechanism that explains the data.

Given that FGF10–FGFR and SHH–PTCH signalling is at the core of the regulatory signalling mechanism governing salivary glands and prostate branching morphogenesis (Fig. 1c, d), it is plausible that a similar ligand–receptor-based Turing-type mechanism controls branching morphogenesis also in these organs. The branching pattern in the kidney differs from that observed in other organs, and there are several important differences in the signalling networks. First, while FGF10 engages in a negative feedback with SHH in the lung, salivary gland, and prostate, GDNF engages in a positive feedback with WNT11[22] (Fig. 1e). Second, while Fgf10 is mainly expressed at a distance from the epithelium[30], Gdnf is initially only expressed in the cap mesenchyme, located adjacent to the ureteric bud epithelium, and from E13.5 also in the stroma[23].

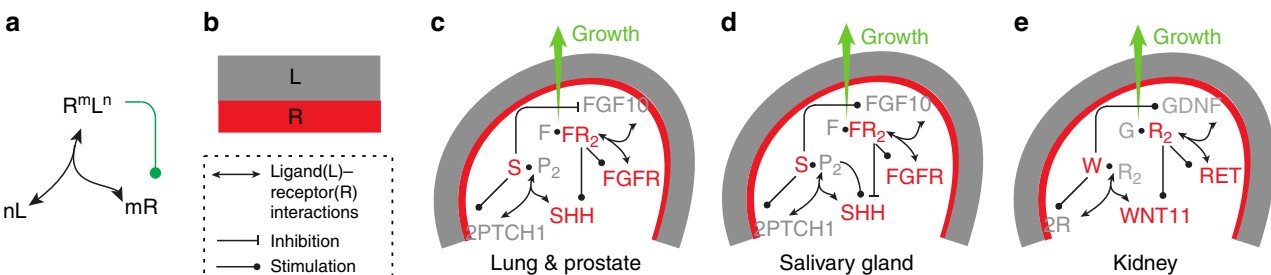

**Fig. 1** Regulatory networks for branching morphogenesis. **a** The network motif shared by all the ligand–receptor pairs that give rise to Turing patterns. **b** Receptors (R) and ligands (L) are expressed in two different tissue layers. The core signalling networks that have been described to regulate branching morphogenesis in **c** lung and prostate, **d** salivary gland, and **e** kidney. **c, d** Fgf10 is expressed in the mesenchyme (grey) and binds to its receptor FGFRIIb in the epithelium (red). FGF10-bound receptors direct the outgrowth of the bud. FGF10 and SHH engage in a negative feedback, in that FGF10 signalling reduces Shh expression in the lung and prostate and increases it in the salivary gland, while SHH signalling increases Fgf10 in the lung and prostate and reduces it in the salivary gland. All ligand–receptor signalling also increases the expression of the receptor. **e** In the kidney, Gdnf is expressed in the mesenchyme (grey) and binds to its receptor RET in the epithelium (red). GDNF signalling induces bud outgrowth and stimulates expression of the receptor Ret and of the secreted ligand Wnt11. WNT11, in turn, increases Gdnf expression in the metanephric mesenchyme. Panels **c–e** were adapted from ref. [3]

We therefore sought to quantitatively compare mechanisms for kidney branching morphogenesis and to explore the impact of the WNT11-dependent positive feedback on the resulting branching pattern. To this end, we obtained movies of cultured embryonic wild type and mutant kidneys, determined the embryonic growth fields, and compared these to the predicted areas of strongest signalling from the different models. Based on the quantitative data, we find that also for the kidney, the ligand–receptor-based Turing-type models recapitulate the outgrowth pattern best. By solving a time-dependent free boundary problem, we further confirm that only the Turing-type ligand–receptor-based models can anticipate and stably mark the point of outgrowth on the deforming domain and thus guide the outgrowth of a domain in the shape of a wild type and mutant developing ureteric bud. Finally, we investigated the impact of the kidney-specific WNT11-dependent positive feedback. We show computationally that this positive feedback permits a denser packing of the developing ureteric buds because the enhanced GDNF availability provides sufficient ligand for both approaching buds. We confirm experimentally that lack of *Wnt11* indeed results in larger inter-bud distances. This observation fits well with previous reports that *Wnt11* mutant mice have smaller kidneys with a lower number of glomeruli[22].

## Results

**Image-based data for a quantitative model selection approach.** It is currently impossible to obtain time-lapse data of kidney branching morphogenesis in utero. We therefore cultured the dissected wild type (Fig. 2a, Supplementary Movie 1) and mutant kidneys (Supplementary Movies 2, 3) under a fluorescent microscope and imaged the branching process once per hour (for details see "Methods"—Kidney cultures and time-lapse imaging). We extracted the shapes of the renal epithelium from the imaging data (Fig. 2b, Supplementary Movies 4–6), and determined the displacement field (Fig. 2c, Supplementary Movies 4–6) (for details see "Methods"—Displacement and growth fields). Growth is strongest at the tips (outward pointing displacement vectors), and thus coincides with the areas of strongest ERK activity (Supplementary Figure 1). In other parts of the domain, we observe shrinkage (inward pointing displacement vectors). As before[43], we do not expect the models to predict the spatio-temporal dynamics of shrinkage, which may occur because of cell migration, deformation, and rearrangements during the budding process, and we therefore remove this information by setting the length of all inward-pointing displacement vectors to zero in our growth field **v** (Fig. 2d).

To solve the models on the embryonic kidney shapes, we require both the epithelial and the mesenchymal layers. The imaging data did not contain information about the mesenchyme. The metanephric mesenchyme, where the ligand *Gdnf* is expressed, can be approximated by an ellipse[1] (Fig. 2e) (for details see "Methods"—Computational domain). The size of the chosen ellipse did not affect the relative performance of the different models, as discussed below. The observed expansion of the *Gdnf* expression domain at later developmental stages[23] should therefore not affect our conclusions.

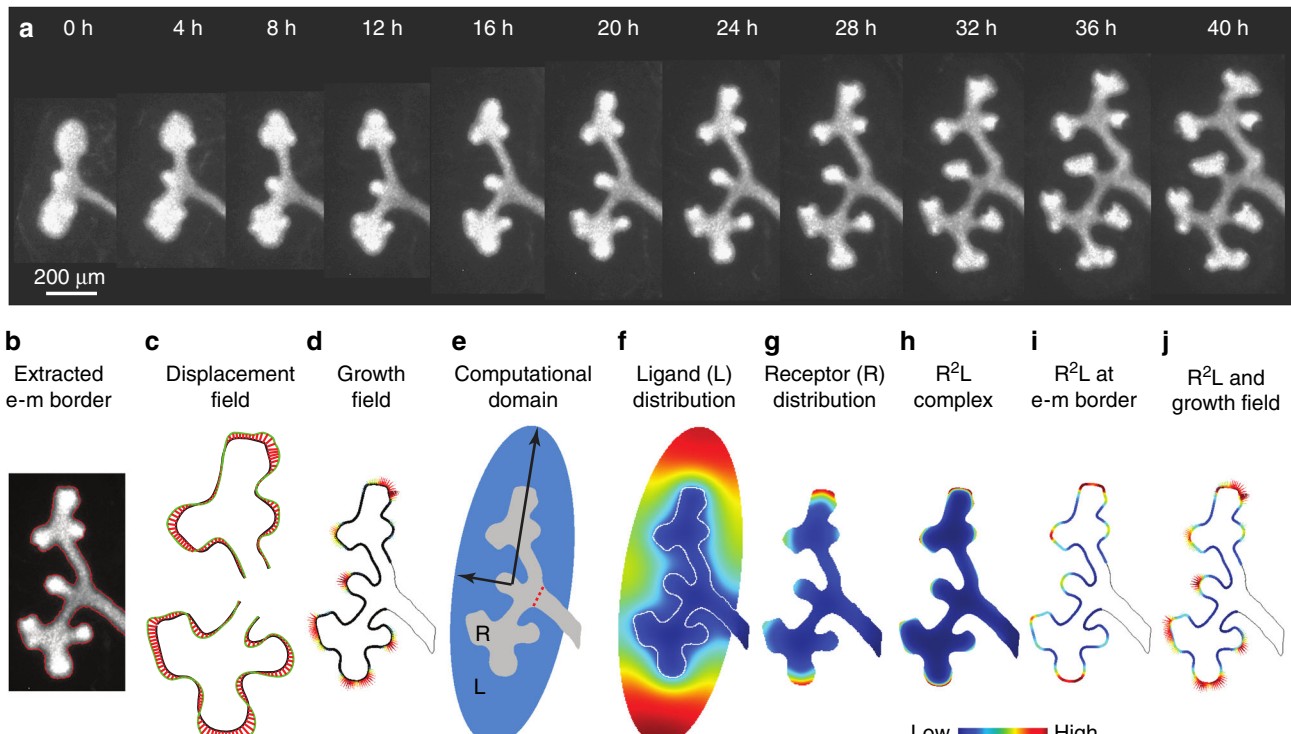

**Fig. 2** The image-based modeling approach. **a** Snapshots from the time-lapse movie of kidney branching morphogenesis at the indicated time points. **b** The red curve marks the extracted border of the epithelium of the ureteric bud. **c** The displacement field (red arrows) between the epithelial border of an earlier (black line) and a later (green line) time point in two consecutive frames. **d** The growth field (red arrows) and the epithelial border (black line) at a given stage. **e** The computational domain comprises the epithelium (grey) and the mesenchyme (blue). The receptor is expressed only in the epithelium, whereas the ligand is expressed only in the mesenchyme. The red dashed line indicates the border of the stalk. **f–h** The computed distribution of the **f** ligand *L*, **g** receptor *R*, and **h** ligand–receptor complex $R^2L$. **i** The computed ligand–receptor signalling strength, $R^2L$, at the border of epithelium and mesenchyme. **j** The predicted ligand–receptor signalling ($R^2L$, solid line) at the epithelium–mesenchyme border and the growth field (vectors). **f–j** The relative strength of the signalling and of the growth field are encoded according to the colourbar

**Alternative models for the control of kidney branching.** Given that the GDNF ligand is secreted close to the epithelium, patterning mechanisms that are based on the distance between the ligand-producing domain and the receptor-expressing domain[30,38,39] cannot apply. This leaves us with only two alternative signalling mechanisms: the ligand–receptor-based Turing model[33,34,42], and the patterning mechanism that is based only on the geometry of the bud[40]. According to the geometry-based mechanism, patterns emerge because the ligand is produced only in one layer (in case of GDNF, the mesenchyme) and diffuses from there to the other layer, where it binds to its receptor (in case of GDNF/RET, the epithelium). While the geometry of the ureteric bud and the restriction of ligand production to the mesenchyme is incorporated also in the Turing model, the Turing mechanism is active only when certain restrictions on the reaction kinetics and parameter sets are met that are discussed below. Accordingly, we can describe both mechanisms by the same set of equations and we can then explore whether the conditions for a Turing mechanism are necessary to reproduce the growth fields in the movies of ureteric bud branching. As derived before[32–34], both mechanisms can be formulated by the following set of partial differential equations (PDEs) for the receptor, $R$, and the ligand, $L$:

$$\text{Epithelium}: \quad \underbrace{\frac{\partial R}{\partial t}}_{\text{time derivative}} = \underbrace{D_{\text{R}}\Delta R}_{\text{diffusion}} + \underbrace{\rho_{\text{R}} - \delta_{\text{R}}R + (\nu - m\mu)R^{\text{m}}L^{\text{n}}}_{\text{biochemical reactions}}$$

$$\text{Epithelium}: \quad \underbrace{\frac{\partial L}{\partial t}}_{\text{time derivative}} = \underbrace{D_{\text{L}}\Delta L}_{\text{diffusion}} - \underbrace{n\mu R^{\text{m}}L^{\text{n}} - \delta_L L}_{\text{biochemical reactions}}$$

$$\text{Mesenchyme}: \quad \underbrace{\frac{\partial L}{\partial t}}_{\text{time derivative}} = \underbrace{D_L \Delta L}_{\text{diffusion}} + \underbrace{\rho_L - \delta_L L}_{\text{biochemical reactions}}$$

$$(1)$$

In the following, we explain and justify this set of equations. Details of the mathematical analysis are described in Supplementary Notes 1. On the left hand side of the equations are the time derivatives describing a local change in molecular concentration. On the right hand side are the diffusion terms, $D_{\text{R}}\Delta R$ and $D_{\text{L}}\Delta L$, and the reaction terms. Here, $D_{\text{i}}$ refers to the diffusion coefficient and $\Delta$ to the Laplace operator, $i = \{R,L\}$. Receptors are produced at the rate $\rho_{\text{R}}$ in the epithelium and ligands are produced at the rate $\rho_{\text{L}}$ in the mesenchyme. The receptors and ligands are turned over independently by linear decay at a low rate $\delta_{\text{i}}$, or can be removed upon complex formation. The stoichiometry of the complex depends on the ligand–receptor pair, and we use the general form of $m$ receptors and $n$ ligands forming a complex, $R^{\text{m}}L^{\text{n}}$. As previously derived[32–34], we can make the quasi-steady state assumption that ligand–receptor binding is much faster than the other reactions and we can thus omit the equation for the ligand–receptor complex. We can then write for the net change in the receptor concentration due to ligand–receptor binding $-m\mu R^m L^n$, and for the net change in ligand concentration due to ligand–receptor binding $-n\mu R^m L^n$. Finally, GDNF–RET signalling has been shown to increase the concentration of RET receptors[11,46]. This reaction is included as $\nu R^m L^n$ in the equation for the receptor dynamics. While the ligand diffuses both in the epithelium and in the mesenchyme, we note that the diffusion of the receptor is restricted to the epithelial layer because diffusion of receptor is restricted to the surface of single epithelial cells. In spite of the restricted receptor diffusion, it is reasonable to solve the model on a continuous domain, because the expected diffusion length within the lifetime of a receptor is smaller than a cell diameter[33,47]. We have previously confirmed that Turing patterns (even with an enlarged Turing parameter space) also occur on cellularized domains[34,48].

A solution of the ligand–receptor-based Turing-type model (Supplementary Table 1: T1) is shown in Fig. 2f–i. Figure 2j shows a comparison between the predicted signalling strength (Fig. 2i) and the growth field that was extracted from the time-lapse data (Fig. 2d).

While Eq. (1) describes both tested models, the Turing model has to meet additional restrictions[32–34,49]. Thus, mathematical analysis via a linear stability analysis (see Supplementary Notes 1 for details) reveals the following necessary conditions for patterns to emerge via a Turing instability in the ligand–receptor model (Eq. (1)): (i) a difference in the diffusion coefficients, $D_L > D_R$; (ii) the presence of a positive feedback of ligand–receptor signalling on receptor abundance, $\nu > m\mu$; and (iii) cooperative interactions of the receptor and the ligand ($m,n \neq 1$ when $m + n = 2$). We therefore tested four alternative models: the complete ligand–receptor-based Turing model (Fig. 3a, Supplementary Table 1, case T1) and three models, in which one of the above conditions was violated (Fig. 3a: T2–T4, Supplementary Table 1: T2–T4). Thus, the model T2 is equivalent to model T1, except that receptors are not upregulated in response to ligand–receptor signalling (Fig. 3a: T2, Supplementary Table 1: T2). The third model, T3, is equivalent to T1, except that the diffusion coefficients of ligand and receptors are set to be equal (Fig. 3a: T3, Supplementary Table 1: T3). The fourth model, T4, is also equivalent to T1, except that the ligand–receptor interaction is not cooperative (Fig. 3a: T4, Supplementary Table 1: T4). Supplementary Table 1 summarises the information about all models that are considered and evaluated in this manuscript.

**Imaging data supports ligand–receptor-based Turing mechanism.** To evaluate the alternative models (Fig. 3a), we sampled several thousand parameter sets for each model from a log-uniform distribution (Supplementary Figure 3). We then compared the predicted normalised concentration of the ligand–receptor complex (Fig. 2i) at the border, $\partial\Omega$, of the epithelium and the mesenchyme,

$$C = \frac{R^2 L}{\max\limits_{\partial\Omega}(R^2 L)}, \quad (2)$$

to the normalised strength of the vector field, $\mathbf{v}$, that represents the growth field extracted from the time-lapse data (Fig. 2f, j),

$$E = \frac{\sqrt{\mathbf{v}^T\mathbf{v}}}{\max\limits_{\partial\Omega}\left(\sqrt{(\mathbf{v}^T\mathbf{v})}\right)}. \quad (3)$$

The choice of normalisation influences only the absolute value of the deviation

$$\Delta = \sqrt{\int\int_{\partial\Omega}(C - E)^2\, d\Omega}, \quad (4)$$

but not the relative value between the different models, and thus the ranking of the models[43]. As we sampled the same parameter sets for all stages $t_i$, a global deviation,

$$\Delta_{\text{g}}(p_j) = \sum_{t_i} \Delta_{t_i}(p_j), \quad (5)$$

could be calculated as the sum of deviations, $\Delta$ (Eq. (4)), at the individual stages. Details of the computational and data analysis

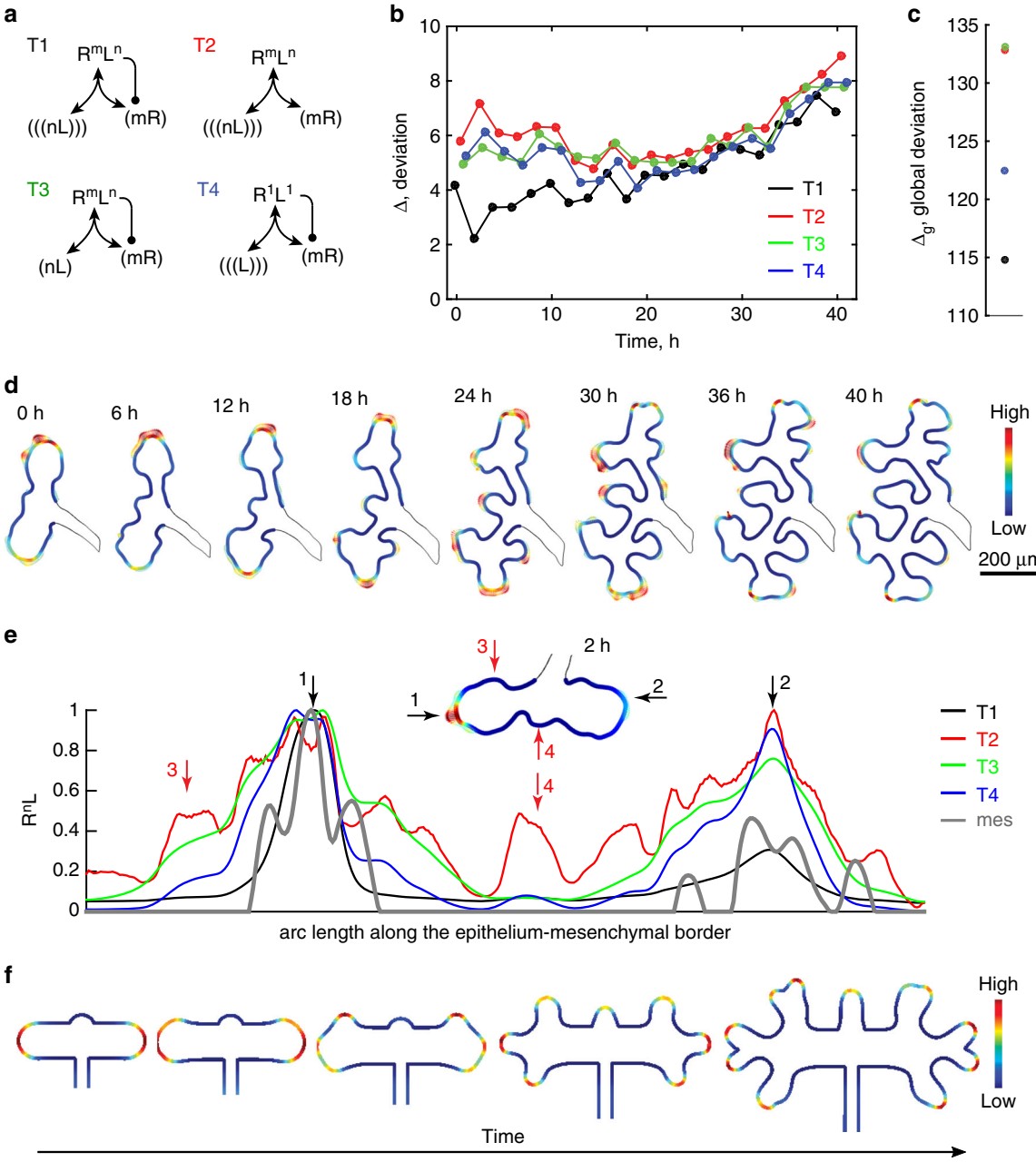

**Fig. 3** Image-based data from wild type kidneys supports a ligand–receptor-based Turing mechanism. **a** Schematic representation of the tested models: (T1) ligand–receptor-based Turing model; (T2) like T1, but without receptor up-regulation; (T3) like T1 but with equal diffusion coefficients for receptors and ligands; (T4) like T1, but with 1:1 stoichiometry of the ligand–receptor complex. A detailed description of the models T1–T4 is provided in Supplementary Table 1. **b**, **c** Minimal deviation between the spatial distribution of signalling strengths $C$ and the experimentally measured growth field, $E$ for **b** each time frame ($\Delta$, Eq. (4)) and **c** globally ($\Delta_g$, Eq. (5)). The colours represent the different models, T1—black, T2—red, T3—green, and T4—blue. **d** The growth areas predicted by the ligand–receptor-based model with the globally optimal parameter set (solid colour) match the growth fields extracted from the experimental data (vectors). The relative strength of the signalling and of the growth field are encoded according to the colourbar. **e** Comparison of the predicted signalling ($R^2L$—colour code for the models T1–T4 as in panels **a**-**c**) and the measured growth fields (grey) along the epithelial-mesenchymal boundary (panel **d**) on an ureteric bud after 2 h of culture. The black arrows indicate points of outgrowth that are correctly predicted by all models. The red arrows highlight examples where the non-Turing models falsely predicted non-existing points of outgrowth. The corresponding ureteric bud shape with the calculated displacement field (vectors) and the best matching signalling field (solid colours) is shown in the inset. The arrows mark the corresponding positions on the boundary. **f** In silico kidney branching morphogenesis. The concentration of the ligand–receptor complex, $R^2L$ (colour code), defines the local growth speed. The resulting computed branching patterns qualitatively recapitulate those observed during branching morphogenesis of explant kidneys. The model parameters are summarised in Supplementary Table 2

procedures are presented in the Methods section; numerical convergence studies are presented in Supplementary Notes 2.

The Turing-type ligand–receptor model (Fig. 3a, Supplementary Table 1, case T1) yielded the smallest deviation, $\Delta$ (Eq. (4)),

between the spatial distribution of signalling strength $C$ and the measured, experimental growth fields $E$ for the vast majority of the analysed time frames (Fig. 3b, black) as well as the smallest global deviation, $\Delta_g$ (Eq. (5)), for the entire time-lapse movie

(Fig. 3c, black) compared to the alternative non-Turing models (Fig. 3a, Supplementary Table 1, case T2–T4). Visual inspection of the simulations with a single globally optimal parameter set (Fig. 3d) and of simulations with different optimal parameter sets for each stage (Supplementary Figure 4) confirms that the Turing-type ligand–receptor model (Fig. 3a: T1) recapitulates the experimentally observed growth fields very well, both on each separate time frame in a series of static computational domains (Fig. 3d) as well as on a continuously growing domain (Supplementary Movie 7). While the quantitative performance, as measured by Δ, between the Turing mechanism and the non-Turing mechanisms may at times appear small, there are important qualitative differences. Most importantly, the non-Turing mechanisms generally locate ligand–receptor signalling to any curved domain (Fig. 3e, Supplementary Figure 5). As a consequence, the non-Turing mechanisms (falsely) locate signalling to the central bud (Fig. 3e, arrow 4) and to an additional curved domain (Fig. 3e, arrow 3) at early time points, even though the central bud emerges primarily due to a shrinkage of the surrounding epithelium and only at the later times due to epithelial outgrowth (Supplementary Figure 6). The Turing mechanism correctly predicts no signalling on those two highly curved domains in those early stages. Furthermore, only the Turing mechanism correctly recapitulates slower growth of the bud on the right hand side (Fig. 3e, arrow 2) compared to the bud on the left hand side (Fig. 3e, arrow 1). The results are independent of the chosen size of the metanephric mesenchyme (Supplementary Figure 7), and we confirmed our results in two further independent datasets (Supplementary Figure 8).

**In silico morphogenesis with a ligand–receptor mechanism**. We next wondered whether any of the mechanisms could actually guide the branching program in silico if the localised signalling triggers the deforming outgrowth of the epithelium. We tested this by solving the different models on a deformable 2D domain with two layers, representing the epithelium and the mesenchyme. The boundary between the two layers was allowed to deform in its normal outward direction with the rate of deformation, i.e. the local growth rate of the epithelium, set to be proportional to the local level of ligand–receptor signalling $R^2L$; the computational details are provided in "Methods"—Deforming outgrowth guided by a signalling model. The ligand–receptor-based Turing-type model (Fig. 3a: T1), unlike any of the non-Turing models (Fig. 3a: T2–T4), qualitatively reproduces the branching patterns (Fig. 3f, Supplementary Figure 9). Thus, kidney explants undergo a trifurcation followed by a bifurcation of newly formed tips (Fig. 3d), and the computational model reproduces this branching pattern (Fig. 3f, Supplementary Movie 8). We note that we have deliberately not tried to reproduce the exact angles of outgrowth because the angles of the cultured kidneys are distorted relative to the angles observed in vivo and the mechanisms that control these are not yet understood and not the focus of this work.

**Validation with genetic and biochemical perturbations**. To further challenge the different branching mechanisms, we used appropriate mutants and biochemical perturbations. We analysed cultured embryonic kidneys from $Fgf10^{-/-}$ (FF) (Fig. 4a–c, Supplementary Movies 2, 5) and $Fgf10^{+/-}$; $Gdnf^{+/-}$; $Spry^{+/-}$ (FGS) (Fig. 4d–f, Supplementary Movies 3, 6) mice as their branching differs from the wildtype. Mathematically, both GDNF and FGF10 signalling can be described by the same equations as they share the same regulatory motif (Fig. 1a), and both ligands (GDNF, FGF10) are expressed in the mesenchyme, while their receptors (RET, FGFR2b) are restricted to the epithelium. $Sprouty1$ is expressed in the epithelium and reduces signalling of the

FGF and other RTK receptors. This affects the strength of signalling and thus the parameter values, but not the structure of the model.

Much as for the wild type, computational parameter screens (Supplementary Figures 10, 11) show that the Turing-type ligand–receptor model yields a smaller deviation, Δ (Eq. (4)) and $Δ_g$ (Eq. (5)), between the spatial distribution of signalling strength $C$ and the experimentally determined growth fields $E$ in the FF and FGS mutants (Fig. 4a, b, d, e, black dots) than any of the alternative ligand–receptor-based non-Turing models (Fig. 4a, b, d, e, green, blue, red dots). This conclusion applies both to the great majority of individual time frames (Fig. 4a, d) and globally (Fig. 4b, e). In spite of the marked differences in the kidney branching patterns in the mutants, visual inspection again confirms that the Turing-type ligand–receptor model (Fig. 3a: T1) recapitulates the growth fields very well and predicts the branch points correctly, both in each frame of a series of static computational domains (Fig. 4c, f) and on the growing domain (Supplementary Movies 9 and 10), using a single globally optimal parameter set. Finally, we confirm that the Turing mechanism can also guide the mutant branching programs in silico. The simulations recapitulate both the branching of the FF mutant kidney explant (Fig. 4c), which undergoes a trifurcation followed by a bifurcation (Fig. 4g, Supplementary Movies 11), and that of the FGS mutant (Fig. 4f), which exhibits substantially reduced branching such that the ureteric tips predominantly elongate and only one tip undergoes branching (Fig. 4h, Supplementary Movies 12). On the contrary, we did not observe branching for any of the alternative models (Fig. 3a: T2–T4), in which the conditions for the Turing mechanism are not fulfilled (Supplementary Figures 12–13). While we cannot exclude that we have just not used the appropriate parameter sets, we have noticed previously that, even though the geometry effect can induce patterns on a curved domain that may look similar to those observed in the branched organs, the geometry effect on its own cannot guide deforming outgrowth[43].

Finally, we biochemically perturbed the organ culture conditions by adding GDNF to the culture, either uniformly or loaded on beads (see Supplementary Notes 3 for details). Uniformly higher GDNF levels result in a widening of the epithelial tissue, both in the experiments (Supplementary Figures 14A, B) and in the simulations (Supplementary Figures 14C, D) as this results in a saturation of signalling-dependent tissue growth. GDNF-loaded beads result in increased local signalling, as evidenced by increased $Etv4$ expression (Supplementary Figure 2), in the local widening of the epithelium due to saturating growth, as well as to increased branching and growth towards the bead, both in the experiments (Supplementary Figure 16A, B) and in silico (Supplementary Figure 16C, D).

We conclude that of the tested models, only the ligand–receptor-based Turing mechanism reproduces and can actively guide the experimentally determined growth patterns in wildtype and mutant kidney cultures, even though the ligand is homogeneously expressed in the mesenchyme, or when added uniformly to the organ culture. This strongly supports the ligand–receptor-based Turing mechanism and excludes other ligand–receptor-based mechanisms.

**A positive feedback reduces the interbud distance**. FGF10/FGF2Rb and GDNF/RET share a common regulatory motif that is compatible with a ligand–receptor-based Turing mechanism, but the gene regulatory network in which they are embedded in lungs and kidneys differs. While FGF10 engages in a negative feedback with SHH in the lung, GDNF engages in a positive feedback with WNT11 in the kidney (Fig. 1e). Thus, GDNF–RET signalling positively

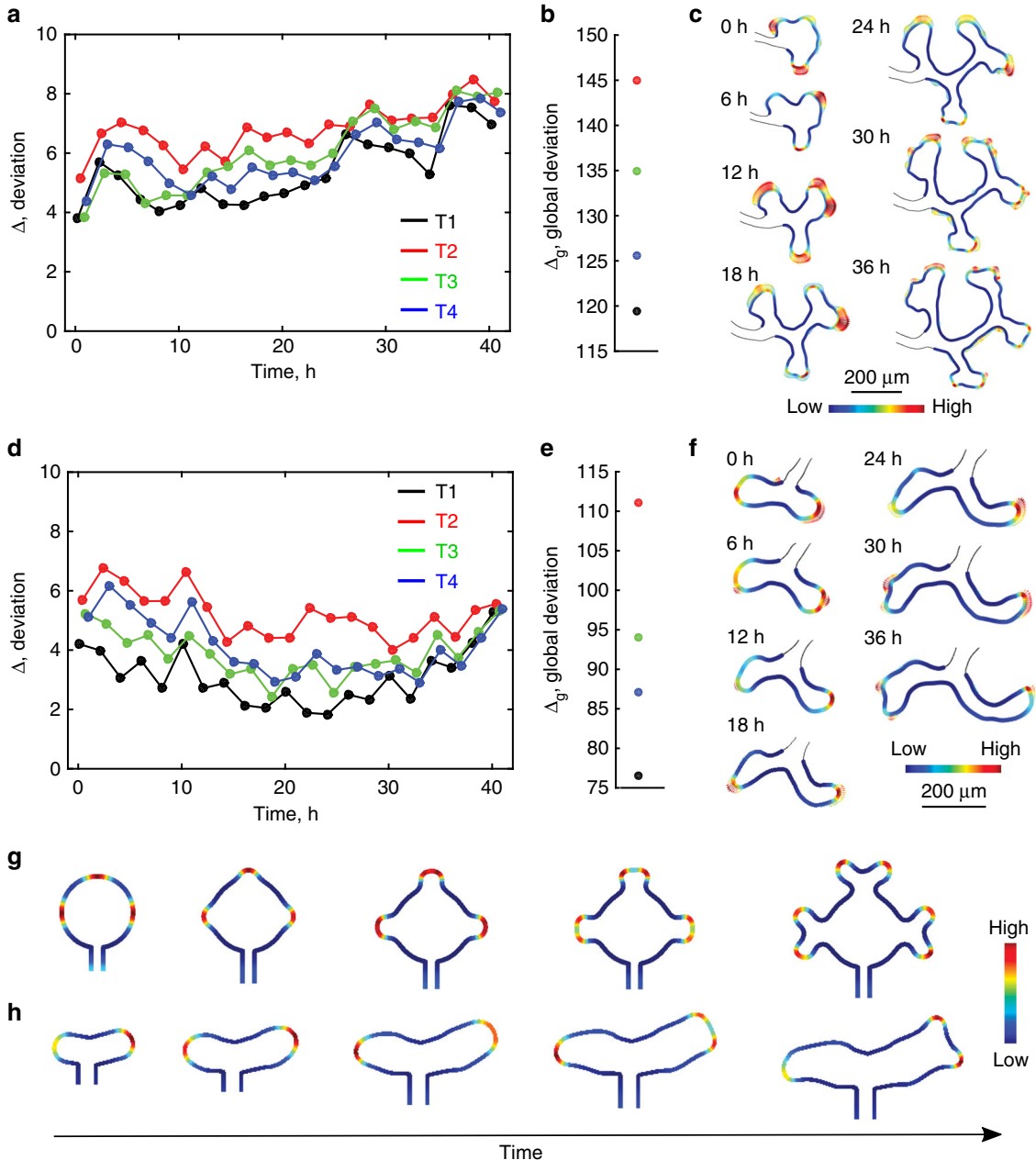

**Fig. 4** Image-based data from mutant kidneys supports a ligand–receptor-based Turing mechanism. **a–f** Analysis of kidney development in **a–c** FF (*Fgf*10$^{-/-}$) and **d–f** FGS (*FGF*10$^{+/-}$;*Gdnf*$^{+/-}$;*Spry*1$^{+/-}$) mutants. Minimal deviation between the spatial distribution of signalling strengths $C$ and the experimentally measured growth field, $E$ for **a**, **d** each time frame ($\Delta$, Eq. (4)) and **b**, **e** globally ($\Delta_g$, Eq. (5)). The colours represent the different models, T1—black, T2—red, T3—green, and T4—blue (see Fig. 3a). **c**, **f** The signalling areas ($R^2L$) predicted by the ligand–receptor-based model (solid colour) match the growth fields extracted from the experimental data (vectors). **g**, **h** In silico branching morphogenesis of mutant kidneys. The concentration of the ligand–receptor complex, $R^2L$ (colour code), defines the local growth speed. The resulting computed branching patterns qualitatively recapitulate those observed during branching morphogenesis of explant kidneys from **g** FF and **h** FGS mutants. The model parameters are summarised in Supplementary Table 2

regulates *Wnt11* expression in the ureteric bud epithelium and WNT11 signalling in turn increases *Gdnf* expression in the mesenchyme, thereby establishing an epithelial–mesenchymal feedback loop[22].

To elucidate the role of the WNT11-dependent positive feedback, we extended the core model (Fig. 3a: T1) to include a second ligand, $L_1$, whose expression is induced by ligand–receptor signalling, $R^2L$, and which in turn increases the production of the ligand $L$ (Fig. 5a, Supplementary Table 1: T5). When we make domain outgrowth dependent on the local $R^2L$ levels (for details

see "Methods"—Deforming outgrowth guided by a signalling model), we notice that the additional positive feedback reduces the minimal interbud distance during the two rounds of branching (Fig. 5b, c, Supplementary Movies 13, 14). On a geometry with two opposing buds at a distance $l_0$, the positive feedback results in increased $R^2L$ levels at the tips and higher ligand expression in the space in between (compare Fig. 5d, e). The positive feedback compensates for ligand sequestration by the neighbouring tips and thereby permits the continuation of outgrowth to lower interbud distances $l_0$ (Fig. 5f, magenta dots),

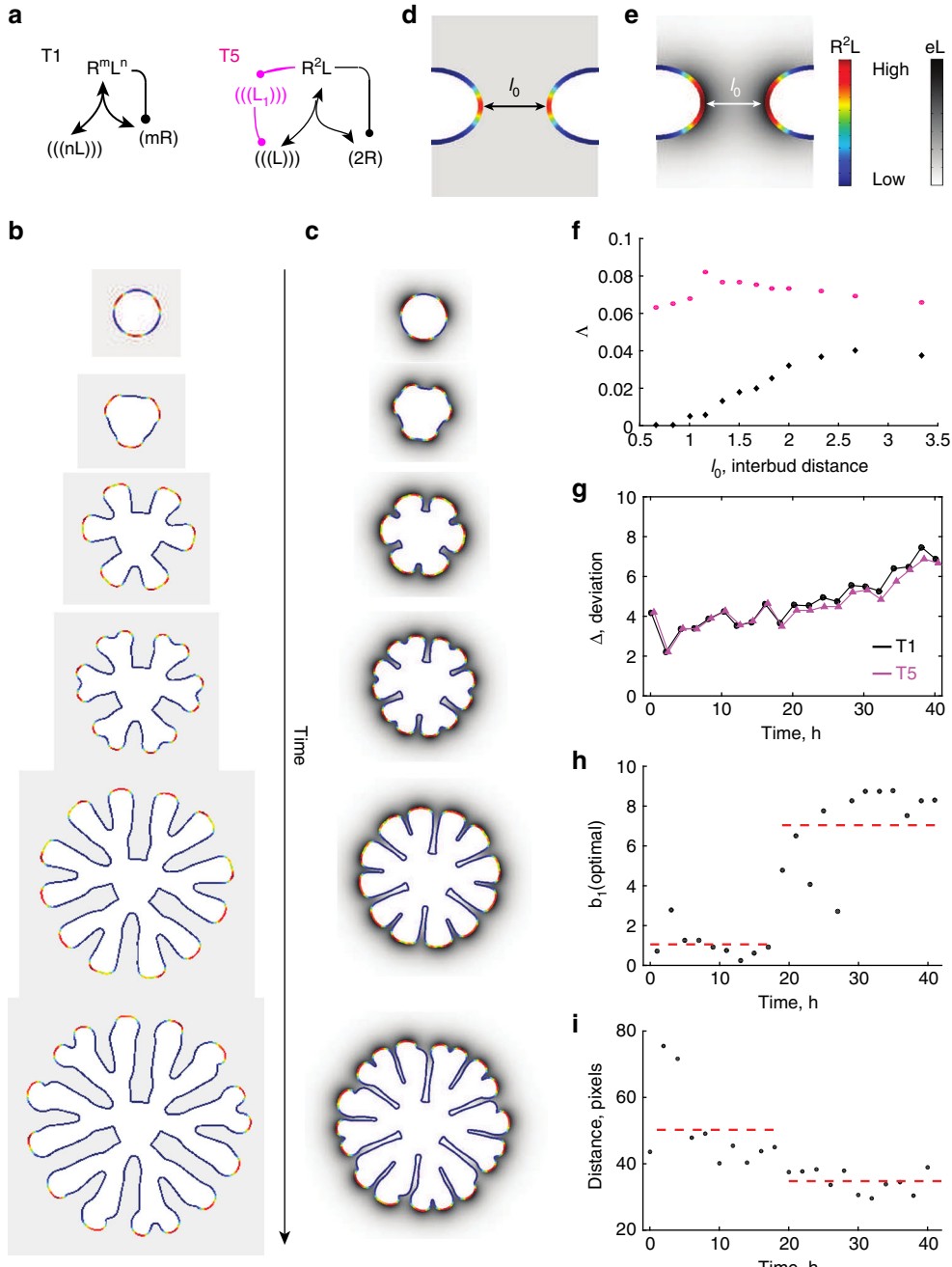

**Fig. 5** A positive feedback regulates the interbud distance. **a** A ligand–receptor-based network motif (T1) and that with an additional positive feedback (in magenta) on ligand production; a detailed model description is provided in Supplementary Table 1: T5. **b, c** Branching patterns simulated on a growing domain with the standard ligand–receptor model, T1 (**b**), and **c** with the model T5 that harbours an additional positive feedback. The parameter values and model details are summarised in Supplementary Table 2. **d, e** Simulated concentration profile of ligand–receptors (rainbow colour code) and *Gdnf* expression (grey scale) for **d** the standard ligand–receptor-based model and **e** for the model T5 with an additional positive feedback. **f** The fraction of sampled parameter sets, $\Lambda$, which lead to the elongation mode of branching for the models T1 (black) and T5 (magenta). **g** Minimal deviation, $\Delta$ (Eq. (4)), between the growth field, $E$, extracted from the experimental data and the predicted spatial distribution of signalling strength $C$ computed with either the complete ligand–receptor-based model (shown in Fig. 3a: T1) (black dots) or the model with an additional positive feedback (shown in **a**: T5) (magenta symbols). **h** The predicted optimal rate of WNT11 production (parameter $b_1$, Supplementary Table 1: T5) is low in the first half, and high in the second half of the culture experiment. The red dashed line depicts the mean values. **i** The average minimal interbud distances in the time-lapse data of cultured ureteric buds are high in the first half, and low in the second half of the culture experiment. The red dashed line depicts the mean values

and increases the overall fraction of parameter sets, $\Lambda$, that support elongating outgrowth of two opposing tips, in particular by enabling elongating growth for lower constitutive ligand production rates $b$ (Supplementary Figure 17; for details on the classification criteria see "Methods"—Pattern classification).

**Wnt11 is required for the dense packing of ureteric tips.** We sought to experimentally test the prediction that the WNT11-dependent feedback reduces the interbud distance in the developing kidney. To this end, we first reanalysed the kidney explant cultures. The positive feedback (Fig. 5g, magenta triangles) results

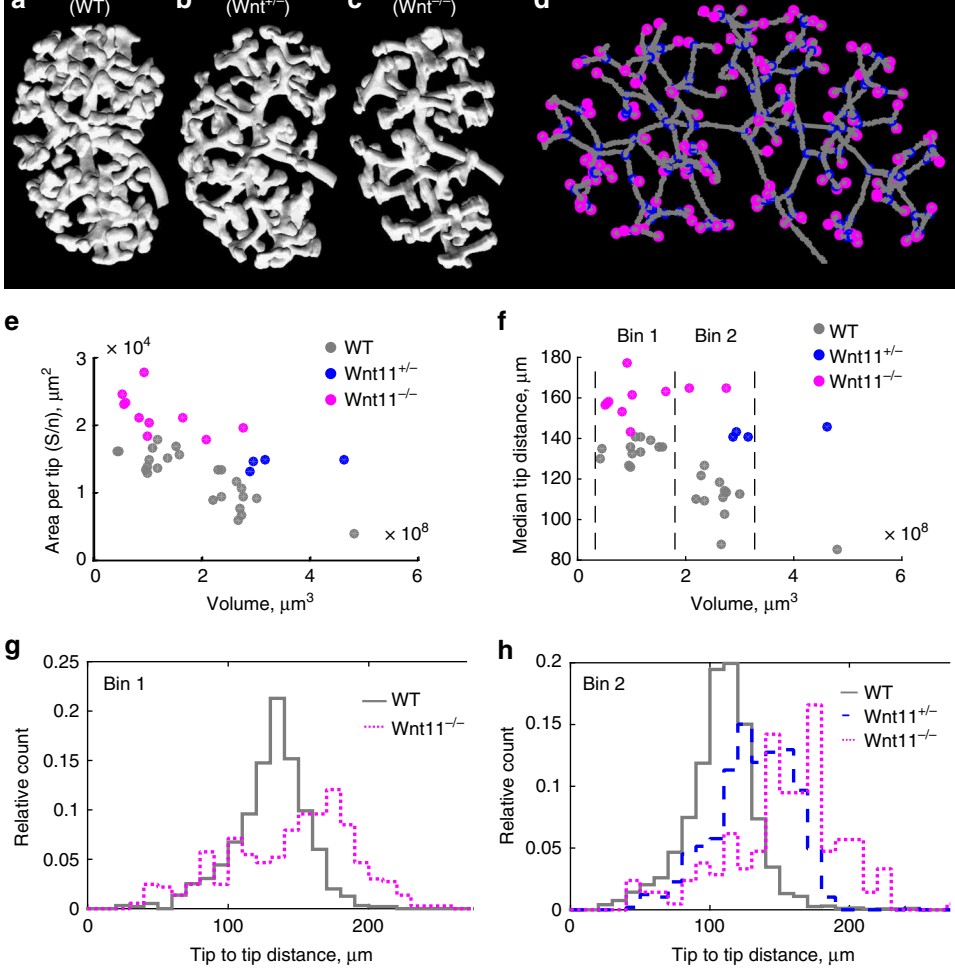

**Fig. 6** WNT11 reduces the interbud distance during kidney branching morphogenesis. **a–c** Representative images of the epithelium from approximately equally sized **a** wild type (E14), **b** *Wnt11*$^{+/-}$ (E13.5), and **c** *Wnt11*$^{-/-}$ (E13.5) embryonic kidneys, as obtained by OPT. **d** Skeletonised renal epithelium (in grey) with epithelial tips marked in magenta and branching points in blue. **e** Area per ureteric bud versus kidney volume in wild type and mutant kidneys. **f** Median tip to tip distance versus kidney volume in wild type and mutant kidneys. **g, h** Distribution of tip to tip distances in **g** small (bin 1) and **h** large (bin 2) wild type and mutant kidneys. Detailed definition of quantitative metrics is available in the Supplementary Methods

in a lower deviation from the data compared to the simple ligand–receptor Turing-type model (Fig. 5g, black dots) only after 20 h of explant culture; until then, the best parameter sets have a low production rate for the additional ligand $L_1$ (denoted $b_1$) (Fig. 5h). Importantly, as predicted (Fig. 5f), the positive feedback becomes relevant exactly at the stage, when the interbud distance becomes smaller (Fig. 5i). Consistent with this, the phenotype of *Wnt11* mutants becomes apparent also only later during kidney development (>E12.5)[22]. In conclusion, the local upregulation of *Gdnf* expression via WNT11 signalling is important only once buds grow towards each other.

To confirm the role of WNT11 in reducing interbud distances, we imaged the epithelium in E13–E14.5 kidneys from wild type (Fig. 6a), *Wnt11*$^{+/-}$ (Fig. 6b), and *Wnt11*$^{-/-}$ (Fig. 6c) mice (for details see "Methods"—Optical projection tomography (OPT) imaging). The 3D renderings of the imaged embryonic kidneys immediately suggest that the density of ureteric tips decreases as the *Wnt11* dosage is reduced. We confirmed this computationally by algorithmically identifying ureteric buds (Fig. 6d) and by evaluating two alternative measures for interbud distance: the surface area per tip, $S/n_{tips}$ (Fig. 6e) and the median tip to tip distance, $t2t\_dist$ (Fig. 6f); details of the definitions and calculations are available in "Methods"—Analysis of the interbud

distance in OPT images of wild type and mutant kidneys. The tip density increases with increasing kidney volume, but according to both measures, wild type kidneys have a higher density of ureteric tips than *Wnt11*-deficient kidneys of comparable size (Fig. 6e, f). Moreover, a reduction in *Wnt11* expression shifts the distributions of tip-to-tip distances to larger values, both in small (Fig. 6g) and large (Fig. 6h) kidneys; the statistical properties of the distributions are summarised in Table 1. This confirms the predictions of the model.

## Discussion

Branching patterns and regulatory networks both differ between branched organs. It is a long-standing question whether branching morphogenesis in the different organs is controlled by a common mechanism despite those differences. We have shown here that the experimentally observed outgrowth pattern during branching morphogenesis of cultured ureteric buds can be recapitulated with the same ligand–receptor-based Turing mechanism (Figs. 3 and 4) as in the lung[33,43], even though the ligand–receptor-based Turing mechanism is based on GDNF–RET interactions in the kidney and on FGF10–FGFRIIb interactions in the lung. All other proposed alternative mechanisms perform worse than the ligand–receptor-based Turing

**Table 1 Interbud distances in wild type and mutant kidneys**

| Genotype | n (samples) | n (tips) | $\bar{x}$, μm | Median, μm | SD[a] | SE[b] |
|---|---|---|---|---|---|---|
| ALL[c] | | | | | | |
| wt[d] | 24 | 4507 | 108.2 | 108.9 | 29.3 | 0.4 |
| Wnt11[+/−][d] | 4 | 701 | 133.6 | 136.2 | 30.2 | 1.1 |
| Wnt11[−/−][d] | 10 | 587 | 147.3 | 155.5 | 43.5 | 1.8 |
| Bin 1[c] | | | | | | |
| wt[d] | 12 | 966 | 130.5 | 133.0 | 28.2 | 0.9 |
| Wnt11[−/−][d] | 8 | 374 | 144.4 | 154.1 | 44.5 | 2.3 |
| Bin 2[c] | | | | | | |
| wt[d] | 11 | 2687 | 108.1 | 109.7 | 25.6 | 0.5 |
| Wnt11[+/−][d] | 3 | 490 | 131.8 | 133.8 | 29.2 | 1.3 |
| Wnt11[−/−][d] | 2 | 213 | 152.3 | 158.0 | 41.2 | 2.8 |

[a]Standard deviation
[b]Standard error
[c]Subset of data analysed: ALL—add data, Bin 1 and Bin 2—subsets of data as indicated in Fig. 6h
[d]Two-sided Welch's t-test rejects equal means hypothesis with $p < 0.001$ at the 5% significance level

models, both in the kidney (Figs. 3 and 4) and in the lung[43]. We further showed that of the tested mechanisms, only the ligand–receptor-based Turing mechanism can guide the outgrowth of an epithelial domain in a way that the same branching patterns emerge in silico as in cultured wild type and mutant ureteric buds (Supplementary Movies 9, 12, 13). We therefore propose that the ligand–receptor-based Turing mechanism constitutes the core regulatory mechanism in both lung and kidney branching morphogenesis.

The branching patterns in lungs and kidneys are rather distinct. We predicted and confirmed experimentally that the additional positive feedback via WNT11[22], which is present only in the kidney, enables the particularly close apposition of buds in the kidney (Figs. 5 and 6). The positive feedback on ligand expression prevents a depletion of ligand when two tips, that both act as a sink for the ligand, come close (Fig. 5d, e). Self-avoidance, and thus the regulation of the closest distance between two ureteric buds, has previously been attributed to BMP7, a repelling signal secreted from the ureteric tips[50]. This is consistent with our findings as genetic evidence points to a negative impact of BMP7 on the GDNF/WNT11 feedback[51].

Turing mechanisms have been proposed to regulate a wide range of patterning phenomena in biology. Some of these have later been shown to be controlled in a different way. It is therefore important to remain cautious. We note, however, that ligand–receptor-based Turing mechanisms have several additional properties that further support them in the context of kidney branching morphogenesis. Most importantly, the Turing mechanism explains why branching is stereotyped only as long as ligands and their receptors are expressed in distinct tissue domains (epithelium and mesenchyme), as observed in all branched organs (Fig. 1a, c–e) as well as in some other developmental systems[52–54]. Turing mechanisms are highly sensitive to noisy initial conditions and can give rise to many different patterns for the same parameter set[43]. Expression of ligands and receptors in distinct tissue layers results in a geometry-induced pre-pattern that biases the Turing mechanism to a single pattern for a given parameter set and thus enables robust stereotyped patterning[43]. As predicted by the Turing mechanism, ectopic expression of Gdnf in the epithelium of the ureteric bud, where its receptor is expressed, results in litters, in which the ureteric branching pattern differs from embryo to embryo[55]. This nonstereotypic patterning in response to homogenous Gdnf expression in the epithelium is difficult to reconcile with alternative mechanisms, but is perfectly consistent with a ligand–receptor-based Turing mechanism.

Simulations of the ligand–receptor-based Turing model recapitulated the branching patterns of both wild type and mutant ureteric buds very well (Figs. 3f and 4g, h), but the branching angles were not captured by the simulations. It is an open question how the branch angles are defined. Related to this is the problem of how buds first emerge once FGF10 and GDNF signalling have marked the points of outgrowth. Results from a number of studies suggest that cell rearrangements and biased cell division shape branching epithelia[20,56–59]. Grafting experiments suggest that also the mesenchyme is an important determinant of the organ-specific branching pattern[60]. Further work is still required to understand how the characteristic shape of different organs emerges from the interplay of epithelial and mesenchymal dynamics, and how mechanical constraints contribute[61–64].

In summary, we have provided quantitative experimental support that a ligand–receptor-based Turing mechanism, implemented via GDNF–RET signalling, specifies the areas of outgrowth during kidney branching morphogenesis. The positive feedback between GDNF and WNT11 enables the dense packing of the ureteric buds. Given its robustness to noise and its flexibility in pattern modulation, the ligand–receptor-based Turing mechanism is likely to be widely used to generate reliable symmetry breaks in biology.

## Methods

**Mouse strains.** The mouse embryonic kidneys depicted in Fig. 2a carried Hoxb7/myrVenus (Tg(Hoxb7-Venus)17Cos/J)[20]. The kidney in Fig. 6 carried Hoxb7/GFP (Tg(Hoxb7-EGFP)33Cos/J) thus expressing the GFP in every UB cell[65]. The Fgf10, Gdnf, and Sprouty1 alleles have been previously described[21]. The Wnt11(tm1a (KOMP)wtsi) was obtained from the International Knockout Mouse Consortium.

All animal experiments conducted in the USA followed PHS policy and guidelines on humane care and use of laboratory animals and were approved and reviewed by the relevant Institutional Animal Care and Use Committees at the University of Southern California and the Columbia University. All procedures conducted at ETHZ were performed in accordance with the ordinance provided by the Canton Basel-Stadt and approved by the veterinary office of the Canton Basel-Stadt, Switzerland (approval number 2777/26711).

**Kidney cultures and time-lapse imaging.** Embryonic kidneys dissected at E11.5 were cultured on Transwell-Clear filters in glass-bottom Petri dishes, in environmentally controlled chambers, as previously described[66]. Time-lapse imaging was performed using a Zeiss Axio Observer Z1 epifluorescence microscope, or a Zeiss LSM 780 confocal microscope. In the experiments with recombinant GDNF, rhGDNF (Cat# 212-GD, R&D) was directly added to the medium at 100 ng/ml. For the bead experiments, Affi-Gel Blue beads (Cat#153-7302, Biorad) were rinsed with PBS and soaked either in rhGDNF at 10 ng/ml or in PBS (control) for 1 h at 37 °C before use. The beads were transferred to a 4-well plate containing PBS. A single soaked bead was then selected and transferred using a P10 micropipette and positioned near an ureteric bud using a tungsten needle. For the whole mount pERK immunostaining, kidneys from E12.5 embryos were dissected in ice-cold

PBS supplemented with PhosSTOP phosphatase inhibitor (Cat# 4906845001 Sigma) and fixed in 4% PFA for 1 h at 4 °C, washed with PBS and blocked in PBS/ 4% BSA (Cat#F9665, Sigma) overnight at 4 °C and stained using a pERK rabbit monoclonal antibody at 1:200 dilution (Cat#4370S, Cell Signaling) for 3 days at 4 °C, washed in PBS during the day and stained using an Alexa Fluor 555 donkey anti-rabbit secondary antibody at 1:500 dilution (Cat#A31572, Molecular Probes) for 3 days at 4 °C, cleared in CUBIC solution[67] and imaged using a Zeiss Z1 light sheet microscope. For *Etv4* in situ hybridization (ISH), the tissue was fixed in 4% PFA overnight at 4 °C after 48 h of live imaging and transwell-filter membranes were cut around samples, which were then processed according to standard whole mount ISH protocols using a digoxigenin-labelled RNA probe for mouse *Etv4*.

**OPT imaging of embryonic kidneys**. Embryonic kidneys were dissected in ice-cold PBS, fixed in 4% PFA 1 h at 4 °C, then washed and stained with an anti-cytokeratin antibody (Sigma) at a 1:250 dilution in 1× PBS supplemented with 5% goat serum (Cat#16210-064, Life Technologies) for 48 h at 4 °C, then washed and stained using an Alexa Fluor 555 goat anti-mouse IgG1 fluorescent secondary antibody (Cat#A21127, Molecular Probes) at a 1:125 dilution in 1× PBS/5% goat serum overnight at 4 °C, then washed again with PBS and processed for OPT as follows. Stained embryonic kidneys were embedded in 1% low melting point agarose, such that the tissue was completely surrounded by agarose. Blocks were trimmed to remove excess agarose, dehydrated in 50% methanol for 6 h, transferred to 100% methanol for 24 h, cleared overnight in a 2:1 mixture of benzyl alcohol/benzyl benzoate and then mounted on a metal OPT magnet, and imaged in a Bioptonics 3001 OPT scanner (Bioptonics, Edinburgh, UK), at maximum resolution of 3.1 µm per pixel zoom. Images were acquired at 0.90° intervals. Post-alignment and 3D reconstruction (filtered back projection method) were performed using NRecon software (Skyscan).

**Image segmentation and border extraction**. We followed the previously reported protocol for image analysis implemented in MATLAB 8.4 (The MathWorks Inc., Natick, MA, 2014)[68]. In brief, the contrast of the images was increased with the built-in MATLAB function imadjust. Next, the images were segmented with a threshold filter (MATLAB function imb2bw). Threshold filters can wrongly assign islands of bright pixels to the kidney epithelium. To eliminate such small islands, we first labelled all separate objects with the MATLAB function bwlabeln and the object with the largest area was selected. We then extracted the border of the epithelium with the MATLAB function bwboundaries (Fig. 2b). Finally the extracted border was approximated with a least-square third order spline (spap2 function in MATLAB).

**Computational domain**. The extracted borders of the kidney epithelium were used to specify the computational domain of the kidney epithelium during the finite element simulations. A typical length scale of the computational domain resembling the ureteric epithelium was 100. The imaging data did not provide the outline of the mesenchymal domain. However, the metanephric mesenchyme, where *Gdnf* is expressed is of elliptic shape and we therefore used an ellipse to represent the computational domain of the mesenchyme. The main axes of the ellipse representing the mesenchyme were set to

$$r_i = 2\alpha * \lambda_i^{1/2} \qquad (6)$$

where, $\lambda_i$ are the eigenvalues of the second moment matrix for the kidney epithelium given by

$$\begin{pmatrix} xx & xy \\ xy & yy \end{pmatrix} \qquad (7)$$

with $xx = \frac{1}{N}\sum_\Omega (x - x_0)^2$, $yy = \frac{1}{N}\sum_\Omega (y - y_0)^2$, and $xy = \frac{1}{N}\sum_\Omega (x - x_0)(y - y_0)$. $x_0$ and $y_0$ are the coordinates of the centre of mass, and $N$ is the number of points in the epithelial border $\partial\Omega$. Note that the points on the boundary were uniformly distributed and the stalk was cut prior to the computation of the centre of mass and of the second moment matrix. The total mass was set to unity and the mass was considered to be uniformly distributed in the epithelium. The size of the ellipse is controlled by $\alpha$, the results given in the main text were calculated with $\alpha = 3$. Computations with different values of $\alpha$ (Supplementary Figure 7) show that while the size of the mesenchyme does affect the absolute values of the deviation, $\Delta$, the relative performance of the alternative models is independent of $\alpha$.

**Displacement and growth fields**. The displacement fields provide the mapping between the geometries of two consecutive stages. The displacement fields between consecutive epithelial boundaries (separated by 2 h) were calculated as the set of vectors that are normal to the epithelial boundary in the current movie frame and that intersect the boundary in the next movie frame[69]. The growth fields were obtained by setting all vectors pointing inward (shrinkage) to zero[43]. Similarly, the displacement fields between two ellipses that represent the mesenchyme for two consecutive stages were calculated.

**Analysis of interbud distances in 2D images**. To calculate the minimal interbud distance, the following procedure was utilised: For every point on the boundary of every bud the minimal distance to the border of all other epithelial buds was determined (distance2curve[70]). If the minimal distance to another bud exceeds the minimal distance to the edge of the image then the point was discarded from the analysis. For details see Supplementary Methods.

**Simulations on the image-based domains**. The PDEs were solved with finite element methods (FEM) as implemented in COMSOL Multiphysics 4.x. COMSOL Multiphysics is a well-established software package and several independent studies confirm that COMSOL provides accurate solutions to reaction-diffusion equations both on constant and growing domains[34,71–74]. The implementation of reaction-diffusion equations on a domain comprising several subdomains has been described previously[75,76].

The models were solved both on static and growing domains. To this end, the computational domains and displacement fields were imported into COMSOL Multiphysics. The deformation of the epithelium was prescribed at the epithelium–mesenchyme border according to the displacement field extracted from the experimental data, and the deformation of the mesenchyme was prescribed at the outer border according to the calculated displacement field between consecutive elliptic shapes. The simulations were carried out iteratively in that the solution after each deformation was mapped to the shape extracted at the next time point (Supplementary Figure 18).

**Convergence of numerical simulations**. The computational models were solved for a wide range of parameter values (Supplementary Figures 3, 10, 11). Convergence studies for the ligand–receptor model T1 show that for a typical parameter set, the deviation $\Delta$ converges to its limiting value as the mesh size decreases (Supplementary Figures 19A, 20A and 21A). In the simulations used to obtain the majority of the results presented in the manuscript, the mesh size was set to 1 on a domain with a typical length scale of 100, resulting in an accuracy for the calculated value of deviation, $\Delta$, of 0.5% or higher. Unlike the ligand–receptor model T1, the alternative models T2 and T4 yielded the minimal values of deviation, $\Delta$, at extreme values for the parameters $a$ and $b$ (Supplementary Figures 3D–F, 10D–F, 11D–F). Convergence studies of the models T2 and T4 showed that as the values of the parameters $a$ and $b$ increase, limiting values of the deviation, $\Delta$, are achieved (Supplementary Figures 19B–D, 20B–D, 21B–D). A fine mesh with element size 0.1 is required to accurately compute the values of deviation, $\Delta$. Sampling of the entire parameter space with such a fine mesh is computationally prohibitive. Therefore, we used it only to calculate the limiting values of the deviation $\Delta$ at high values of the parameters $a$ and $b$. These limiting values of deviation $\Delta$ are depicted by the dashed red line in Supplementary Figures 3D–F, 10D–F, 11D–F. The limiting value of the deviation $\Delta$ yields the minimal possible value of the deviation $\Delta$ for models T2 and T4 and therefore is depicted in Figs. 3 and 4. Further details of the convergence studies are provided in Supplementary Notes 2: Convergence and Accuracy of the Computational Method.

**Deforming outgrowth guided by a signalling model**. To test whether the ligand–receptor-based Turing type model can lead branching morphogenesis, we solved a series of models (Supplementary Tables 1,2) on a deforming domain. The deformation at the epithelium–mesenchyme border was set normal to the epithelial–mesenchyme border and proportional to the local concentration such that the velocity field was given by $\bar{v} = \bar{n}v_g f(R^2L)$, $\bar{n}$ is the outward vector pointing normal to the epithelium–mesenchyme border and $f(R^2L)$ is provided in Supplementary Table 2. The outer layer of the mesenchyme was set to expand normal to the surface with a constant speed $v_g$. The growth was modelled in the quasi-steady-state limit, such that the same branching pattern is obtained also with any lower rate $v_g$. The typical value of the growth rate was $v_g = 0.04$. To maintain a high quality of the finite element mesh during the simulations, the computational domain was re-meshed at regular time intervals. The mesh elements were chosen to be sufficiently fine so that further refinements did not influence the observed branching pattern.

**Pattern classification**. Patterns simulated on the domain depicted in Fig. 5b, c were classified as those corresponding to the elongation mode of branching if patterns observed on the epithelium–mesenchyme border of both tips satisfied the following criteria:

(1)   Pattern has substantial amplitude: $\max(R^2L)/\min(R^2L) > 5$;
(2)   A number of peaks in $R^2L$ distribution is equal to 1;
(3)   Peak $R^2L$ is located in the centre of the bud, within 10% accuracy.

As a measure for the size of the parameter space where elongation mode of branching is observed we defined $\Lambda = \frac{n_{el}}{n_t}$, where $n_{el}$ is the number of sampled parameter sets, which lead to the elongation mode of branching, and $n_t$ is the total number of sampled parameter sets. The value of $\Lambda$ lies between 0 and 1, where 0 indicates that an elongation mode of branching is not observed in the model and 1 indicates that all the sampled parameter sets support the elongation mode of branching.

**Analysis of epithelial bud density in OPT kidney images**. To quantify the epithelial bud density in 3D images of wild type and *Wnt11* mutant kidneys, we implemented an image analysis pipeline in Fiji 2.0[77] and MATLAB 8.4 (The MathWorks Inc., Natick, MA, 2014). The pipeline included the following steps: image de-noising, local thresholding (Supplementary Figures 22A), skeletonisation, and skeleton voxel classification (Supplementary Figures 22B, C). Details of image analysis procedure are available in Supplementary Methods.

Epithelial bud density was quantified according to two independent metrics: the surface area per tip, $S/n_{tips}$, and the median tip to tip distance, $t2t\_dist$ (Supplementary Figures 22D–F). Here, the surface area per tip, $S/n_{tips}$ is defined as the surface area of the smallest ellipsoid that encloses all ureteric tips divided by the total number of ureteric tips. The median tip to tip distance, $t2t\_dist$, is defined as the median of the tip to tip distances in the immediate neighbourhood of a given ureteric tip (Supplementary Figures 22D). Samples were grouped according to their volume; the volume was determined as the smallest ellipsoid that encloses the kidney epithelium.

**Reporting summary**. Further information on experimental design is available in the Nature Research Reporting Summary linked to this article.

**Code availability**. MATLAB and ImageJ code for image processing and analysis will be made available on request.

## Data availability
The data that support the findings of this study are available from the corresponding author upon reasonable request.

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

## Acknowledgements

The authors are grateful to Frank Costantini for providing mice and reagents and for discussions, to Rolf Zeller for access to the OPT scanner, to Peter Whitney for technical assistance, and to Srivathsan Adivarahan for conducting some preliminary computational analysis. The authors thank members of the Iber group, in particular Simon Tanaka, for comments on the manuscript and for discussions. O.M. is grateful for an EMBO Short-Term Fellowship to visit the laboratory of A.P.M. Work in A.P.M.'s laboratory was supported by a grant from the NIH (DK054364).

## Author contributions

D.I. and D.M. conceived the study. D.M. carried out all computational analysis. O.M., C.L. and L.C. carried out all experimental work in the laboratories of Frank Costantini, A.P.M. and D.I. All authors approved the final version of the manuscript.

## Additional information

**Competing interests:** The authors declare no competing interests, except for D.M., who is employed by Bayer AG. The opinions expressed in this manuscript are those of the authors and do not necessarily reflect the views of the employer.

