## [Peer Review File · Nature Communications]

Reviewers' comments:

Reviewer #1 (Remarks to the Author):

OVERALL: This paper uses a combination of image analysis (of real wet-lab data) and computer modelling to test various alternative models, and present a very plausible Turing-based model for epithelial branching in the developing kidney. The paper is very timely, and following as it does from similar work on the lung, it makes a very important point – that anatomists' instinct that different branching systems are at some level the same 'under the hood' stands up mathematically. The results will be significant both to developmental biology and to evolutionary studies. Overall, the work is very well done and well-written. I have made fairly extensive suggestions for corrections below but this is mainly to improve the precision of the writing and to address some points that embryologists will expect to be acknowledged. The length of my suggestions should not be interpreted negatively – this is a very good and important paper and when it is corrected and published it may be a classic in the field.

Title: inclusion of the word 'core' seems unnecessary; it tries to force a particular view of development that is not proved by the data. The title would work very well without this word, and would be more true to what is presented.

Abstract: the abstract contains the very strong claim that a previous publication showed that 'only' a ligand-receptor based Turing mechanism can account for branching in the lung. Are the current authors absolutely certain that the previous publication really did prove that, rather than showing the weaker conclusion that, '**of a finite range of possibilities tested**', only the Turing one worked?

Introduction – first paragraph – the glands mentioned *contain* branched structures (the current text says 'are' – most of the glands are not branched structures as seen from the outside at the gross anatomy level). In the same paragraph 'branching program' carries a lot of metaphorical baggage: is the use of 'program' (rather than, say, 'mechanism' really justified here?).

Introduction – parag 2 – do you need that sentence about control of branching probably being best studied in lung or kidney? It strikes me as a way to alienate those who study salivary glands, prostate, mammary gland etc without having any positive impact on your paper. Same paragraph: you keep writing 'core regulatory mechanism' – again, your use of 'core' brings with it a lot of metaphorical baggage (forcing readers to buy into a view of development you do not justify, and probably have more sense than to try to justify). If all you mean is 'important' or 'necessary', please avoid 'core'. (This comment applies to later parts of the introduction and the figure legend for fig 1. If you insist on core, please include a proper introduction with citations about why you are using this view of developmental mechanisms).

Introduction – parag 2 - *important* - the Ret knockout phenotype is being exaggerated in this paragraph and it is a critical point in the assumptions of the model. The text seems to state that branching cannot take place in the absence of Ret. This is not the case. If you look at Schuchardt et al. (1994) – PubMed PMID: 8114940 – you will note that some kidneys of the knockout mice did form and managed to show the first branching events, described in text and in pictures in that paper. It is critical that the authors of this current paper do not mislead their readers on this point (far too many authors of reviews have glossed over it and have created a false impression about Ret being essential for all kidney branching). A complete model must show an ability for some branching in the absence of Ret too – if it does not, then the model needs to be acknowledged as an approximation with the question of how Ret-/- mice can show any branching being left as an open question (with this issue flagged clearly in the Discussion). I am not arguing here that the model presented is not useful, just that its 'fit' should not be exaggerated. Similarly, GDNF -/-

mice do not show a 100% loss of ureteric bud branching (see Pichel et al 1996 PMID 8657307 – see near the end of the penultimate paragraph on p73 of the pdf of their paper). This may be because of redundancy with the other Ret ligand Neurturin, also present in the kidney (PMID 10322636). Same paragraph – that GDNF induces epithelial outgrowth and that it is a chemoattractant for branches were first shown by Kirsi Sainio (PMID 9374404), a paper that should be cited (by all means also cite the papers you do, but the Sainio one is arguably closer to the natural situation you model because of the way it was done, using beads in organ culture): this comment applies to the introduction and also to the paragraph in the results under the heading under equation 2.

Introduction, paragraph at the top of page 4. The kidney is complicated, because it starts with the UB branching in the mesenchyme and soon progresses to having the mesenchyme organized as Six2+ 'caps' around the UB tips and stromal progenitors – which have recently been shown to be the main source of GDNF – outside the caps. It is reasonable for the current paper to model only the initial situation and to ignore this change (dealing with that can be their next paper!), but the introduction has to acknowledge the complication – without the acknowledgement the authors may seem naive. Rather than give a lot of references here, I'll suggest the corresponding author should chat to their co-author McMahon, who definitely knows about this having been a co-author of the paper that rediscovered the nephrogenic caps (first described about 100 years ago). Same paragraph – please don't use 'upregulates', another word that brings a whole lot of semantic baggage, when the data justify only 'increases'. Actually, this problem occurs a lot later in the paper – what do you mean by 'upregulate' that goes beyond 'increase'? (it's the 'regulate' part that seems problematic – sounds like adding more/ extra levels of regulation: I know the word is widely used but that does not mean it is right).

Introduction, last paragraph, please use 'wild type' and not just 'wt' (partly for flow, and partly because kidney people often see 'Wilms tumour' when they see the letters 'wt'). In this paragraph, we return to the same problem flagged for the abstract – the claim that 'only Turing-type models can...' when the truth is that 'of the models tested, only the...' (it's a big difference – tomorrow someone may come up with a brand new model that also works).

Results – paragraph 2 under 'alternate models' - the comment that "the distance between the ligand-producing domain and the receptor- expressing domain [34, 40, 41] can therefore not apply to the kidney." is true at the very beginning (UB in unorganized MM) but arguably not by the time the Six2+ caps have formed, because the GDNF, it turns out, is made mainly by the stromal progenitors outside the caps and therefore there is a gap between the ligand-producing domain and the receptor-expressing domain. The movies used for the analyses, and the time-periods modelled (judged by numbers of branches in the final tree), span the two periods, the disorganized MM one and the capped one. Assuming the authors take my advice and briefly address this complication in the Introduction, this section of the Results needs to restrict the quoted comment to the simple unorganized MM condition, and refer the reader back to the Introduction for clarification. Immediately after the quoted line is the statement that only two models remain: this again carries an assumption that all possible other models have been eliminated (rather than the truth that the others in the finite set imagined by the authors can be eliminated).

The method of extracting data from time-lapse movies was done well and explained brilliantly in both text and figures. Round of applause...

Equation 3 – another round of applause about how well the text labels explain this!

Results paragraph under equation 3: the authors correctly spend time justifying a critical assumption, that the cell surface of the UB can be modelled continuously without regard for cell boundaries, with respect to receptor diffusion. They say this is justified by the fact that receptor half-life is about the same as time to diffuse one cell diameter, and refer to ref 37 for this. This reference is another similar paper, which is about a different receptor (Ptc) and which anyway

does not seem to cite a reference for the Ptc half-life given (670s). Really we need a half-life for Ret here; Ptc is not relevant. The only source I could find on Ret's half life (PMID 20444924; different cell type) states Ret half-life as some 4 hours, which is far, far longer than that given in ref 37 for Ptc, and far too long to justify ignoring cell boundaries. If the truth is that cell boundaries were ignored to avoid a computational nightmare, the authors should be honest about this and not invoke assumptions about half-life that seem, from the paper just mentioned, to be false.

Page 15 end of paragraph 1 – Wnt11 is important only in later stages. Interesting – again there may be a link with the switch to cap mesenchyme organization, ignored in these models. I'm not sure what the authors can do/say about this, but I mention it in case it triggers an interesting thought for them. There is no mandatory correction associated with this point.

Videos: these are of high quality and well presented. The raw data ones ought to have a scale bar (if this is not possible to add, maybe state the whole frame size, in um, in the video figure legend – that will be adequate).

As Nat Com supports reviewers signing, I will do so, congratulating the authors again on an interesting paper.

Jamie Davies, Edinburgh.

Reviewer #2 (Remarks to the Author):

Denis Menshykau et al. manuscript entitled "Image-Based Modeling of Kidney Branching Morphogenesis reveals a core GDNF-RET based Turing-type Mechanism and a pattern-modulating WNT11 Feedback" targets one of the key questions for kidney morphogenesis namely the mode of molecular and signaling actions by which the mammalian kidney obtains its shape. This question has been targeted mostly experimentally and recently also via systematic imaging approaches. However, the field has lacked the in depth mathematical analysis, simulations and combination of the generated experimental data to the modelling approaches. This paper now is an opening for this direction as a technical documentation.

Based on molecular genetic evidence where individual genes were genetically inactivated in vivo phenotypes that also relate to the changes in the kidney ureteric epithelial bifurcation have been identified. Based on the fact that these assays have been grounded to the gene targeting approach the evidence acquired in the field is rather solid. In the current, this data and the capacities to culture and identify the exact details how the kidney acquires its shape was used. Also, the kidney ureteric tree 3D structure can be depicted by using tomographic technologies such as the optical projection tomography (OPT).

These above indicated technological issues establish the embryonic kidney as a suitable organ for the current work, namely aims to model the process and acquire more detailed mathematical based ways to evaluate the process with Image-Based Modeling of Kidney Branching Morphogenesis illustrations.

In the paper two types of models were tested how these calculations would offer an output that would have relevance to the mode of development of the kidney organ, the bench layout. The ways how the models were tested appears well enough completed and no major problems appears in the calculations and the generated tests were well conducted.

Collectively the novelty in the work is to target the validity of the Turing model behind dynamics

of the epithelial ureteric bud branching morphogenesis, the major tests being were 1) the Turing type mechanism (ligand-receptor based Turing-model) and 2) the patterning mechanism that is based only on the geometry of the bud. The conclusions drawn are that the Turing-type mechanism based predictions seem to fit better in to the actual calculated values, thus better than the competing model tested. The noted mistakes in the predictive models were smaller in the case of the Turing type of mechanism. As it appears the work and the evaluations are done accurately enough.

Some specific issues:

The problematic pages are 7-10. The word order appears not fluent and for example in the text there appears many time words such as 'since, thus, and via' (i.e. redundancy). The equation symbols usually should be explained directly after the equations, even if written in results, or then just have another supplementary section for all of the equations. The referee that the supplementary section is not at all of its part very representative. The paper appears rather technical paper Some polishing needs to be done. As for the computer vision and segmentation this appears adequately done.

In summary the paper targets an important question in kindey development, namely the dynamics if the epithelial branching process and tests the classic models in their value to model it and to offer parameters for further studies.

Reviewer #3 (Remarks to the Author):

In this paper, Menshykau and colleagues use computational modeling to argue that a Turing-type mechanism, involving GDNF, Ret, and Wnt11, regulates the branching morphogenesis of the ureteric bud. A set of reaction-diffusion equations is used to simulate the interactions between GDNF, which is expressed in the kidney mesenchyme, and its receptor Ret, which is expressed in the ureteric epithelium. The authors compare spatial patterns of the bound GDNF-Ret complex in their simulations to experimental measurements of the motion of the epithelial border during the branching morphogenesis of cultured kidney explants. These data are then used to claim that a Turing-type mechanism regulates the overall branching pattern in both wild-type and Wnt11-/- mutant kidneys.

Although the scientific questions the authors address are interesting, Menshykau and colleagues do not provide sufficient experimental evidence to support their assertion, based on the results of their computational model, that Turing-type patterns control branching morphogenesis in the kidney.

Their computational model predicts specific distributions of Ret-signaling within the ureteric epithelium and of GDNF expression within the mesenchyme, but these predictions are not compared with direct experimental measurements of either. Instead, the authors use geometric and kinematic data, gathered from cultured and fixed embryonic kidneys explants, to make inferences about the validity of their model. They assume, for instance, that the tracked motion of the epithelium during branching morphogenesis can be used to measure epithelial "growth." This assumption conflates "growth" with the complex tissue deformations that underlie branching morphogenesis, deformations that may be caused in part by epithelial proliferation (although the authors do not report any experimental measurements of proliferation or "growth") but are also likely to be the result of other mechanical forces, such as cytoskeletal contraction or exogenous fluid forces.

The authors have chosen to test their model by comparing simulated patterns of Ret-activation with this kinematic measure of epithelial "growth." Based on the data provided in the manuscript, it is not clear why this comparison is a valid one. Clear experimental data that shows the

emergence of Turing-type patterns in embryonic kidney explants, such as immunofluorescence imaging of activated downstream components in the Ret-pathway, or in situ hybridization measurements of GDNF expression, is needed to properly test the validity of the model.

Still, even with a more robust comparison between model and experiment, it would be important that the authors demonstrate their computational model can predict abnormal branching patterns when GDNF-Ret signaling is disrupted. They cite a study showing that GDNF-loaded beads, embedded within the mesenchyme, lead to abnormal kidney branching morphogenesis and induce the formation of ectopic buds (Pichel et al., 1996). How might a focal source of GDNF in the mesenchyme perturb the hypothesized Turing-type patterns? And how might experimental evidence of these disrupted patterns compare to simulated patterns in the authors' computational model?

Without clear experimental evidence showing the emergence of Turing-type patterns in embryonic kidneys during both normal and disrupted branching morphogenesis, I cannot recommend this manuscript for publication in Nature Communications.

Response to Reviewers' comments:

Reviewer #1 (Remarks to the Author):

OVERALL: This paper uses a combination of image analysis (of real wet-lab data) and computer modelling to test various alternative models, and present a very plausible Turing-based model for epithelial branching in the developing kidney. The paper is very timely, and following as it does from similar work on the lung, it makes a very important point – that anatomists' instinct that different branching systems are at some level the same 'under the hood' stands up mathematically. The results will be significant both to developmental biology and to evolutionary studies. Overall, the work is very well done and well-written. I have made fairly extensive suggestions for corrections below but this is mainly to improve the precision of the writing and to address some points that embryologists will expect to be acknowledged. The length of my suggestions should not be interpreted negatively – this is a very good and important paper and when it is corrected and published it may be a classic in the field.

We thank the reviewer for his positive evaluation.

Title: inclusion of the word 'core' seems unnecessary; it tries to force a particular view of development that is not proved by the data. The title would work very well without this word, and would be more true to what is presented.

As suggested by the referee, we removed the word 'core' from the title.

Abstract: the abstract contains the very strong claim that a previous publication showed that 'only' a ligand-receptor based Turing mechanism can account for branching in the lung. Are the current authors absolutely certain that the previous publication really did prove that, rather than showing the weaker conclusion that, ****of a finite range of possibilities tested**, only the Turing one worked?**

We indeed meant to say what the referee suggests. We now hope to make this clearer by saying “of all **previously proposed signaling-based mechanisms**”:

We have shown **in earlier work** that of all **previously proposed signaling-based mechanisms** only a ligand-receptor based Turing mechanism based on FGF10 and SHH can quantitatively recapitulate the branching patterns in the lung.

Introduction – first paragraph – the glands mentioned *contain* branched structures (the current text says 'are' – most of the glands are not branched structures as seen from the outside at the gross anatomy level).

We have changed the wording as suggested by the referee:

Many organs in higher animals, such as the lung, the kidney and the glands **contain** complex branched structures, which emerge via a process called branching morphogenesis.

In the same paragraph 'branching program' carries a lot of metaphorical baggage: is the use of 'program' (rather than, say, 'mechanism' really justified here?).

We have changed the wording to:

The capacity of these organs and the kidney, in particular, to perform their physiological function depends on the **emergence of the correct branching structure**.

Introduction – parag 2 – do you need that sentence about control of branching probably being best studied in lung or kidney? It strikes me as a way to alienate those who study salivary glands, prostate, mammary gland etc without having any positive impact on your paper.

We certainly did not intend to offend anyone, and we have changed the wording to:

The control of branching morphogenesis **is well-studied** in the lung and in the kidney.

Same paragraph: you keep writing 'core regulatory mechanism' – again, your use of 'core' brings with it a lot of metaphorical baggage (forcing readers to buy into a view of development you do not justify, and probably have more sense than to try to justify). If all you mean is 'important' or 'necessary', please avoid 'core'. (This comment applies to later parts of the introduction and the figure legend for fig 1. If you insist on core, please include a proper introduction with citations about why you are using this view of developmental mechanisms).

We are using core for the regulatory parts that are necessary to observe branching morphogenesis. Several other signaling pathways modulate the branching patterns, but are not key to their emergence. It is not entirely clear to us what other metaphorical baggage this term would carry. We now explicitly specify our definition of core in the context of this paper:

In summary, FGF10 and to a slightly lesser extent SHH signalling are necessary for lung branching morphogenesis, while GDNF signalling is necessary for kidney branching morphogenesis, though, to a certain extent, redundant pathways can compensate for loss of GDNF signalling. Interference with other signalling pathways alters, but does not prevent lung and kidney branching morphogenesis. In the following, we will refer to these necessary signalling pathways as "core" regulatory pathways.

Introduction – parag 2 - *important* - the Ret knockout phenotype is being exaggerated in this paragraph and it is a critical point in the assumptions of the model. The text seems to state that branching cannot take place in the absence of Ret. This is not the case. If you look at Schuchardt et al. (1994) – PubMed PMID: 8114940 – you will note that some kidneys of the knockout mice did form and managed to show the first branching events, described in text and in pictures in that paper. It is critical that the authors of this current paper do not mislead their readers on this point (far too many authors of reviews have glossed over it and have created a false impression about Ret being essential for all kidney branching). A complete model must show an ability for some branching in the absence of Ret too – if it does not, then the model needs to be acknowledged as an approximation with the question of how Ret-/- mice can show any branching being left as an open question (with this issue flagged clearly in the Discussion). I am not arguing here that the model presented is not useful, just that its 'fit' should not be exaggerated.

Similarly, GDNF -/- mice do not show a 100% loss of ureteric bud branching (see Pichel et al 1996 PMID 8657307 – see near the end of the penultimate paragraph on p73 of the pdf of their paper).

This may be because of redundancy with the other Ret ligand Neurturin, also present in the kidney (PMID 10322636).

This is a very interesting point – thank you very much!! Paragraph 2 has been modified the following way:

Kidney branching morphogenesis is largely inhibited in the absence of Glial-derived neurotrophic factor (GDNF) signalling, though some of the null mutants still develop rudimentary kidneys, suggesting that additional pathways contribute to UB branching [3,4,20-22]. These likely include neurturin, which like GDNF binds to the epithelial receptor RET and its co-receptor GFR α 1, as well as FGF10, which binds to the epithelial receptor FGFR2b [20,23,24]. Both, the RET and the FGF receptor tyrosine kinase pathways, converge on ERK signalling and the transcription factors ETV4/5 [20]. Removal of *Sprouty1*, an intracellular inhibitor of ERK signalling, from a *Ret* or *Gdnf* null mutant background restores UB branching and kidney development [20,24].

Same paragraph – that GDNF induces epithelial outgrowth and that it is a chemoattractant for branches were first shown by Kirsi Sainio (PMID 9374404), a paper that should be cited (by all means also cite the papers you do, but the Sainio one is arguably closer to the natural situation you model because of the way it was done, using beads in organ culture): this comment applies to the introduction and also to the paragraph in the results under the heading under equation 2.

Thank you very much – we have added the reference to the text.

Introduction, paragraph at the top of page 4. The kidney is complicated, because it starts with the UB branching in the mesenchyme and soon progresses to having the mesenchyme organized as Six2+ 'caps' around the UB tips and stromal progenitors – which have recently been shown to be the main source of GDNF – outside the caps. It is reasonable for the current paper to model only the initial situation and to ignore this change (dealing with that can be their next paper!), but the introduction has to acknowledge the complication – without the acknowledgement the authors may seem naive. Rather than give a lot of references here, I'll suggest the corresponding author should chat to their co-author McMahon, who definitely knows about this having been a co-author of the paper that rediscovered the nephrogenic caps (first described about 100 years ago).

Thank you. We have now discuss the dynamics of the *Gdnf* expression pattern:

Importantly, while *Fgf10* is mainly expressed in the submesothelial lung mesenchyme, i.e. at a distance from the epithelium [30], *Gdnf* is initially expressed in the cap mesenchyme, which is located directly adjacent to the ureteric bud epithelium. From E13.5, *Gdnf* expression has also been observed in the stroma, i.e. outside the cap mesenchyme [31].

Same paragraph – please don't use 'upregulates', another word that brings a whole lot of semantic baggage, when the data justify only 'increases'. Actually, this problem occurs a lot later in the paper – what do you mean by 'upregulate' that goes beyond 'increase'? (it's the 'regulate' part that seems problematic – sounds like adding more/ extra levels of regulation: I know the word is widely used but that does not mean it is right).

We did mean to indicate a regulatory interaction in that GDNF-RET signaling regulates the expression of *Wnt11* in a way that it results in increased *Wnt11* expression. But we are also happy to use the word “increase” instead of “upregulate”.

In this paragraph, we return to the same problem flagged for the abstract – the claim that 'only Turing-type models can...' when the truth is that 'of the models tested, only the...' (it's a big difference – tomorrow someone may come up with a brand new model that also works).

We have changed this to:

By combining imaging data with computational modelling, we challenged the different signalling-based mechanisms for lung branching morphogenesis with embryonic data and showed **that of all previously proposed signaling-based mechanisms** only a ligand-receptor based Turing-type mechanism based on FGF10-FGFR and SHH-PTCH signalling in combination with a tissue specific expression of ligands and receptors [39,40,46] correctly recapitulates the experimentally observed areas of growth during lung branching morphogenesis [47].

Results – paragraph 2 under 'alternate models' - the comment that “the distance between the ligand-producing domain and the receptor- expressing domain [34, 40, 41] can therefore not apply to the kidney.” is true at the very beginning (UB in unorganized MM) but arguably not by the time the Six2+ caps have formed, because the GDNF, it turns out, is made mainly by the stromal progenitors outside the caps and therefore there is a gap between the ligand-producing domain and the receptor-expressing domain. The movies used for the analyses, and the time-periods modelled (judged by numbers of branches in the final tree), span the two periods, the disorganized MM one and the capped one. Assuming the authors take my advice and briefly address this complication in the Introduction, this section of the Results needs to restrict the quoted comment to the simple unorganized MM condition, and refer the reader back to the Introduction for clarification. Immediately after the quoted line is the statement that only two models remain: this again carries an assumption that all possible other models have been eliminated (rather than the truth that te others in the finite set imagined by the authors can be eliminated).

The important point for the model is that, unlike *Fgf10* in the lung, *Gdnf* is expressed close to the epithelium. As long as there is substantial *Gdnf* expression close to the epithelium, as distance-based mechanism cannot apply – even if there is additional expression at a greater distance from the epithelium. A recent paper discussed the stromal expression of *Gdnf* in length:

Magella, B., et al. (2018). "Cross-platform single cell analysis of kidney development shows stromal cells express Gdnf." *Dev Biol* 434(1): 36-47.

Using three different methods for scRNASeq, the authors consistently observe *Gdnf* expression in E14.5 kidney stroma cells, and they confirm their finding with E13.5 images from the Allen Brain atlas and their own ISH analysis at an unspecified stage (presumably E14.5). However, importantly, they still observe *Gdnf* expression also in cap mesenchyme cells, in particular with the Fluidigm system, the most sensitive of the three systems. This result would thus still be inconsistent with a distance-based mechanism. In our manuscript, we are focusing on the regulation of the early stages of kidney branching morphogenesis (the cultures are carried out with E11.5 kidneys). In summary, we do not believe that this concern applies here. We now state this explicitly:

Gdnf is expressed throughout the metanephric mesenchyme adjacent to the epithelium [29], **and in later stages also in the stroma** [31].

Moreover, the size of the mesenchyme domain does not affect the ranking of the different models. We now emphasize this in the text:

To solve the models, we require both the epithelial and the mesenchymal layers. The imaging data did not contain information about the mesenchyme. The metanephric mesenchyme, where the ligand *Gdnf*

is expressed, is approximately of elliptic shape [3], and accordingly, we approximated the shape of the metanephric mesenchyme with an ellipse (Fig 2E) (for details see Material and Methods → Analysis of Time-Lapse Data → Computational Domain). We note that the size of the chosen ellipse did not affect the relative performance of the different models as discussed below. **The observed expansion of the *Gdnf* expression domain at later developmental stages (Magella et al, 2018) should therefore not affect our conclusions.**

The method of extracting data from time-lapse movies was done well and explained brilliantly in both text and figures. Round of applause...

Equation 3 – another round of applause about how well the text labels explain this!

Thank you!!

Results paragraph under equation 3: the authors correctly spend time justifying a critical assumption, that the cell surface of the UB can be modelled continuously without regard for cell boundaries, with respect to receptor diffusion. They say this is justified by the fact that receptor half-life is about the same as time to diffuse one cell diameter, and refer to ref 37 for this. This reference is another similar paper, which is about a different receptor (*Ptc*) and which anyway does not seem to cite a reference for the *Ptc* half-life given (670s). Really we need a half-life for *Ret* here; *Ptc* is not relevant. The only source I could find on *Ret*'s half life (PMID 20444924; different cell type) states *Ret* half-life as some 4 hours, which is far, far longer than that given in ref 37 for *Ptc*, and far too long to justify ignoring cell boundaries. If the truth is that cell boundaries were ignored to avoid a computational nightmare, the authors should be honest about this and not invoke assumptions about half-life that seem, from the paper just mentioned, to be false.

The relevant time here is the time to internalization. The reported half-life for RET51 is 20 minutes, which is within the range of our argument. We have now included a reference to the RET paper.

Richardson, D. S., et al. (2012). "Alternative splicing results in RET isoforms with distinct trafficking properties." *Mol Biol Cell* **23**(19): 3838-3850.

We note that Turing-type patterns can also be observed on cellular domains, e.g. Menshykau et al PLoS Comp Biol 2012 and Kurics et al Phys Rev E 2014.

Page 15 end of paragraph 1 – *Wnt11* is important only in later stages. Interesting – again there may be a link with the switch to cap mesenchyme organization, ignored in these models. I'm not sure what the authors can do/say about this, but I mention it in case it triggers an interesting thought for them. There is no mandatory correction associated with this point.

There may well be such link – possibly in that WNT11 can induce *Gdnf* expression at a longer distance. But this would be rather speculative, and we would rather prefer to not enter this discussion, given the lack of data.

Videos: these are of high quality and well presented. The raw data ones ought to have a scale bar (if this is not possible to add, maybe state the whole frame size, in um, in the video figure legend – that will be adequate).

This has been added.

As Nat Com supports reviewers signing, I will do so, congratulating the authors again on an interesting paper.

Jamie Davies, Edinburgh.

Thank you, Jamie!! This has been an extraordinary thoughtful and helpful review – thank you very much!!

Reviewer #2 (Remarks to the Author):

Denis Menshykau et al. manuscript entitled “Image-Based Modeling of Kidney Branching Morphogenesis reveals a core GDNF-RET based Turing-type Mechanism and a pattern-modulating WNT11 Feedback” targets one of the key questions for kidney morphogenesis namely the mode of molecular and signaling actions by which the mammalian kidney obtains its shape. This question has been targeted mostly experimentally and recently also via systematic imaging approaches. However, the field has lacked the in depth mathematical analysis, simulations and combination of the generated experimental data to the modelling approaches. This paper now is an opening for this direction as a technical documentation.

Based on molecular genetic evidence where individual genes were genetically inactivated in vivo phenotypes that also relate to the changes in the kidney ureteric epithelial bifurcation have been identified. Based on the fact that these assays have been grounded to the gene targeting approach the evidence acquired in the field is rather solid. In the current, this data and the capacities to culture and identify the exact details how the kidney acquires its shape was used. Also, the kidney ureteric tree 3D structure can be depicted by using tomographic technologies such as the optical projection tomography (OPT).

These above indicated technological issues establish the embryonic kidney as a suitable organ for the current work, namely aims to model the process and acquire more detailed mathematical based ways to evaluate the process with Image-Based Modeling of Kidney Branching Morphogenesis illustrations.

In the paper two types of models were tested how these calculations would offer an output that would have relevance to the mode of development of the kidney organ, the benched layout. The ways how the models were tested appears well enough completed and no major problems appears in the calculations and the generated tests were well conducted.

Collectively the novelty in the work is to target the validity of the Turing model behind dynamics of the epithelial ureteric bud branching morphogenesis, the major tests being were 1) the Turing type mechanism (ligand-receptor based Turing-model) and 2) the patterning mechanism that is based only on the geometry of the bud. The conclusions drawn are that the Turing-type mechanism based predictions seem to fit better in to the actual calculated values, thus better than the competing model tested. The noted mistakes in the predictive models were smaller in the case of the Turing type of mechanism. As it appears the work and the evaluations are done accurately enough.

Thank you very much.

Some specific issues:

The problematic pages are 7-10. The word order appears not fluent and for example in the text there appears many time words such as 'since, thus, and via' (i.e. redundancy).

We have re-read and restructured the first part of the Results section in the hope that it is now easier to read. As non-native speakers, we may not fully meet the referee's expectations as to the elegance of the writing, but we hope that the writing is now clear and easy to follow.

The equation symbols usually should be explained directly after the equations, even if written in results, or then just have another supplementary section for all of the equations.

We agree with the referee and we have checked this again: all symbols are defined immediately before or after they are used. In case of Equation 3, it takes an entire paragraph to define them, but this is because we need to explain what they are exactly. To make this clearer, we now added the following sentence immediately after the equations:

In the following, we explain and justify this set of equations. Details of the mathematical analysis are described in the Supplementary Text S-1.

Moreover, we have moved the discussion of continuous versus discrete cellular domains to the end of the paragraph.

The referee that the supplementary section is not at all of its part very representative.

We have carefully revised the supplementary material: we have fixed several textual issues and we have tried to make several figures clearer, in particular, Fig S-7, S-8, S-12, S-13

The paper appears rather technical paper Some polishing needs to be done.

In response to the reviewer's comment, we have now revised the main text, figures and supplementary material. In particular, we have reorganized the Results section, and we have split the old Figure 7 into two Figures, and we have introduced additional panels to facilitate the understanding of the quantitative evaluation of tip to tip distance. We hope that the revised manuscript is an easier read.

As for the computer vision and segmentation this appears adequately done.

Thank you.

In summary the paper targets an important question in kindey development, namely the dynamics if the epithelial branching process and tests the classic models in their value to model it and to offer parameters for further studies.

Thank you.

Reviewer #3 (Remarks to the Author):

In this paper, Menshykau and colleagues use computational modeling to argue that a Turing-type mechanism, involving GDNF, Ret, and Wnt11, regulates the branching morphogenesis of the ureteric bud. A set of reaction-diffusion equations is used to simulate the interactions between GDNF, which is expressed in the kidney mesenchyme, and its receptor Ret, which is expressed in the ureteric epithelium. The authors compare spatial patterns of the bound GDNF-Ret complex in their simulations to experimental measurements of the motion of the epithelial border during the branching morphogenesis of cultured kidney explants. These data are then used to claim that a Turing-type mechanism regulates the overall branching pattern in both wild-type and Wnt11/- mutant kidneys.

Although the scientific questions the authors address are interesting, Menshykau and colleagues do not provide sufficient experimental evidence to support their assertion, based on the results of their computational model, that Turing-type patterns control branching morphogenesis in the kidney.

Their computational model predicts specific distributions of Ret-signaling within the ureteric epithelium and of GDNF expression within the mesenchyme, but these predictions are not compared with direct experimental measurements of either. Instead, the authors use geometric and kinematic data, gathered from cultured and fixed embryonic kidneys explants, to make inferences about the validity of their model. They assume, for instance, that the tracked motion of the epithelium during branching morphogenesis can be used to measure epithelial "growth." This assumption conflates "growth" with the complex tissue deformations that underlie branching morphogenesis, deformations that may be caused in part by epithelial proliferation (although the authors do not report any experimental measurements of proliferation or "growth") but are also likely to be the result of other mechanical forces, such as cytoskeletal contraction or exogenous fluid forces.

The authors have chosen to test their model by comparing simulated patterns of Ret-activation with this kinematic measure of epithelial "growth." Based on the data provided in the manuscript, it is not clear why this comparison is a valid one. Clear experimental data that shows the emergence of Turing-type patterns in embryonic kidney explants, such as immunofluorescence imaging of activated downstream components in the Ret-pathway, or in situ hybridization measurements of GDNF expression, is needed to properly test the validity of the model.

All of these aspects are already well established, i.e. the *Gdnf* expression domains are available in many publications and in the Allen Brain Atlas (see also the point by referee 1 on this). Moreover, phosphorylated ERK and *Etv4/5* expression, which are both downstream of GDNF-RET signaling, have previously already been shown to become concentrated at the tips of ureteric buds (Ihermann-Hella, A., et al. (2014). "Mitogen-activated protein kinase (MAPK) pathway regulates branching by remodeling epithelial cell adhesion." *PLoS Genet* **10**(3): e1004193. and Lu, B. C., et al. (2009). "Etv4 and Etv5 are required downstream of GDNF and Ret for kidney branching morphogenesis." *Nat Genet* **41**(12): 1295-1302.). We now cite this work also in the Results section, and provide additional experimental datasets as Supplementary Figure 1 & 2, as requested.

The following text and supplementary figures were added to the beginning of the Results section:

GDNF-RET signalling induces renal epithelial outgrowth and acts as a renal epithelial chemoattractant [23–25]. Phosphorylated ERK and *Etv4* expression, which are both downstream of GDNF signalling, are concentrated in the tips of ureteric buds [49, 50] (Figs S-1, S-2A). Moreover, GDNF-loaded beads increase *Etv4* expression locally [50] (Fig. S-2B). Accordingly, the correct model needs

to explain how GDNF signalling becomes concentrated at the epithelial tips.

Still, even with a more robust comparison between model and experiment, it would be important that the authors demonstrate their computational model can predict abnormal branching patterns when GDNF-Ret signaling is disrupted. They cite a study showing that GDNF-loaded beads, embedded within the mesenchyme, lead to abnormal kidney branching morphogenesis and induce the formation of ectopic buds (Pichel et al., 1996). How might a focal source of GDNF in the mesenchyme perturb the hypothesized Turing-type patterns? And how might experimental evidence of these disrupted patterns compare to simulated patterns in the authors' computational model?

We emphasize that already in the previous submission, we included mutants with abnormal branching patterns. To make this clearer, we have now split the previous validation section and discuss the mutants separately. In addition, we have now performed the requested culture experiments where we added GDNF either homogeneously or locally as GDNF-soaked beads. We indeed confirm that the observed alterations are consistent with the proposed ligand-receptor based Turing mechanism. The following text has been added to the manuscript:

Finally, we biochemically perturbed the organ culture conditions. Uniform addition of GDNF ligand to the embryonic kidney culture results in a widening of the buds (Fig. S-14A,B). This effect can be qualitatively recapitulated in silico (Figure S-14C,D), and is a consequence of a saturation of signalling-dependent tissue growth at receptor-saturating GDNF concentrations. We next studied the effect of a locally restricted GDNF ligand source in the form of a GDNF-loaded bead (Figs S-15, S-16). The effect of a local source of ligand, in this case GDNF, on ligand-receptor based Turing patterns can be understood by recalling that the instability that arises from the ligand-receptor interactions has a particular wave-length, that manifests itself as the distance between branching points. This wavelength is dependent on the parameter values, including the ligand production and turn-over rates. A local source of ligand (GDNF) enhances the local availability of the ligand and therefore modulates the wavelength. Simulations on a simplified, static two-layer domain, that idealises the epithelial and mesenchymal layers, show that at a sufficiently high concentration of the ligand in the bead, an additional spot with high ligand-receptor signalling can emerge either directly near the local ligand source (Fig S-15A-C), or a peak can split into two peaks (Figure S-15D-F). The exact effect of an additional ligand source is difficult to anticipate due to the nonlinear nature of the system and must therefore be studied computationally. GDNF-loaded beads have both of the above described effects in growing embryonic kidney cultures: a) a widening of the epithelial tissue in the proximity of the bead due to saturation of growth, and b) increased branching and growth toward the bead (Fig S-16A,B). The simulations recapitulate the observed behaviour qualitatively (Fig S-16C,D). The effect of the GDNF-loaded bead was confirmed by its effect on *Etv4* expression. The GDNF-loaded bead indeed increased *Etv4* expression levels in the epithelial tips in proximity of the GDNF source (Fig S-2), therefore pointing to a local increase in RET signalling.

Without clear experimental evidence showing the emergence of Turing-type patterns in embryonic kidneys during both normal and disrupted branching morphogenesis, I cannot recommend this manuscript for publication in Nature Communications.

The ultimate proof of a Turing mechanism requires the quantitative measurements of all reaction and diffusion rates in the tissue of interest. This is infeasible to date. But our work provides strong support for a ligand-receptor based Turing mechanism in kidney branching morphogenesis. As experimental techniques develop, proposed mechanisms will have to be re-evaluated – but this applies to any scientific domain. We very much hope that the now provided additional experimental

and computational evidence will dispel the reviewer's concerns regarding the relevance of a Turing mechanism for kidney branching morphogenesis.

REVIEWERS' COMMENTS:

Reviewer #1 (Remarks to the Author):

I am completely happy with the way that the authors have responded to the points I have raised.

I think this is an important paper that will be well cited for some years to come.

Reviewer #4 (Remarks to the Author):

First of all I am asked to assay whether the authors have successfully replied to the issues raised by the referee 3, so I limit my answer to this specific issue. The answer is formally yes but in general no - the authors have answered the examples raised by the referee 3 but the these examples seem to be insufficient for the convincing evidences for the existence of Turing mechanism. There are various possible mechanisms of branching morphogenesis (mechanics, diffusion limited growth, L-system etc) and they authors does not provide crucial evidences to discriminate these models. In other system, James Sharpe group have provided huge amount of experimental data beginning from screening of candidate molecules and in depth mathematical analyses to show skeletal pattern formation is based on Turing mechanism (two Science papers) but the issue is still controversial. Comparing to that better example, I do not think referee 3 is satisfied with this list of superficial similarities.

As a general rule of review process the paper should be published but I feel the problem is still open.

Response to Referees:

Reviewer #1 (Remarks to the Author):

I am completely happy with the way that the authors have responded to the points I have raised.

I think this is an important paper that will be well cited for some years to come.

Thanks, Jamie!!

Reviewer #4 (Remarks to the Author):

First of all I am asked to assay whether the authors have successfully replied to the issues raised by the referee 3, so I limit my answer to this specific issue. The answer is formally yes but in general no - the authors have answered the examples raised by the referee 3 but these examples seem to be insufficient for the convincing evidences for the existence of Turing mechanism. There are various possible mechanisms of branching morphogenesis (mechanics, diffusion limited growth, L-system etc) and they authors does not provide crucial evidences to discriminate these models. In other system, James Sharpe group have provided huge amount of experimental data beginning from screening of candidate molecules and in depth mathematical analyses to show skeletal pattern formation is based on Turing mechanism (two Science papers) but the issue is still controversial. Comparing to that better example, I do not think referee 3 is satisfied with this list of superficial similarities.

As a general rule of review process the paper should be published but I feel the problem is still open.

First of all, we want to thank the referee for taking the time to evaluate another person's referee report. We fully agree with the referee that in no system final proof has yet been provided that a Turing mechanism controls the patterning. However, we do not agree that other previously proposed mechanisms might apply instead for the kidney and were just ignored by us. The particular system that we consider has been studied in great detail, and many details are known regarding molecular mechanisms and mutant phenotypes. This provides a sufficient number of constraints that we indeed can and do exclude other previously proposed patterning mechanism. This, of course, does not exclude the possibility that a different, not yet proposed mechanism applies – but this is true for any scientific study. In the following, we comment on the ones that the referee mentioned in particular.

We first note that the stereotypic nature of the branching process is an important feature of lung and kidney development. As a result, stochastic mechanisms cannot apply as we stress in the paper. This excludes many possible mechanisms such as viscous fingering and many mechanical instabilities as we discuss in our review "Iber, D. and D. Menshykau (2013). "The control of branching morphogenesis." Open biology 3(9): 130088-130088.".

L-systems correspond to rule-based modelling, and virtually any rule can be implemented. We are not aware of any rules that would have been proposed based on known molecular mechanisms in the kidney that we could test with our imaging data.

The distance-based mechanism that we discuss in detail in the paper and reject based on image-based data corresponds to diffusion-limited growth as we discuss in detail in our review "Iber, D. and D. Menshykau (2013). "The control of branching morphogenesis." Open biology 3(9): 130088-130088." Given the space constraints, we do not see how to discuss this further in this paper.

Finally, since the first submission of this paper, additional patterning mechanisms based on mechanical feedbacks have been proposed that have not yet been tested for lung and kidney development, and we now explain why they would not apply, at least in their current form:

Recently, a coupling of morphogen dynamics and morphogen-induced shape changes has been shown to result in patterning (Brinkmann et al, 2018). Here, the shape changes had to result in locally enhanced morphogen production. The latter would not fit with the situation in lungs and kidneys where the morphogens are produced in a different tissue (mesenchyme) from the one that deforms (epithelium). Moreover, tissue stretching as predicted from mechanical models did not coincide with measured branch points (George et al, 2015). While more complicated feedback architectures that involve morphogens and tissue mechanics could be explored, we note that the ligand-receptor based Turing mechanism represents a parsimonious mechanism that explains the data.

We want to emphasize that the fact that we exclude all these mechanisms for lung and kidney branching morphogenesis does, by no means, mean that we exclude alternative possibilities that have not yet been discussed / discovered. We only state that the proposed mechanism is the only one that we are aware of that can explain the molecular data – and not only the general branching architecture.

We hope that this addresses the concerns of the reviewer.